# AMPylation targets the rate-limiting step of BiP's ATPase cycle for its functional inactivation

Steffen Preissler*, Lukas Rohland[†], Yahui Yan, Ruming Chen[‡], Randy J Read, David Ron*

Cambridge Institute for Medical Research, University of Cambridge, Cambridge, United Kingdom

**Abstract** The endoplasmic reticulum (ER)-localized Hsp70 chaperone BiP contributes to protein folding homeostasis by engaging unfolded client proteins in a process that is tightly coupled to ATP binding and hydrolysis. The inverse correlation between BiP AMPylation and the burden of unfolded ER proteins suggests a post-translational mechanism for adjusting BiP's activity to changing levels of ER stress, but the underlying molecular details are unexplored. We present biochemical and crystallographic studies indicating that irrespective of the identity of the bound nucleotide AMPylation biases BiP towards a conformation normally attained by the ATP-bound chaperone. AMPylation does not affect the interaction between BiP and J-protein co-factors but appears to allosterically impair J protein-stimulated ATP-hydrolysis, resulting in the inability of modified BiP to attain high affinity for its substrates. These findings suggest a molecular mechanism by which AMPylation serves as a switch to inactivate BiP, limiting its interactions with substrates whilst conserving ATP.

DOI: https://doi.org/10.7554/eLife.29428.001

*For correspondence:
sp693@cam.ac.uk (SP);
dr360@medschl.cam.ac.uk (DR)

Present address: [†]Center for Molecular Biology of Heidelberg University, Heidelberg, Germany; [‡]Greenase Biosynthesis Ltd, Xiamen, China

## Introduction

Compartment-specific chaperones contribute substantially to folding of newly synthesized polypeptides and to protein turnover and thereby facilitate maintenance of proteome integrity (*Bukau et al., 2006*). The abundant Hsp70-type chaperone BiP (or Grp78) is a central component of the chaperone repertoire of the endoplasmic reticulum – the gateway to the secretory pathway of eukaryotic cells. BiP mRNA and protein levels have long been known to respond to changes in the burden of unfolded secretory proteins; induction of BiP-encoding mRNA being a hallmark of the unfolded protein response (UPR) (*Chang et al., 1989*; *Kozutsumi et al., 1988*). However, in the secretory pathway fluctuations in the unfolded protein load occur on time scales that are too short to be accommodated solely by slow and costly antagonistic regulation of BiP levels by transcription and protein degradation. Therefore, post-translational mechanisms for rapidly adjusting the level of active BiP to the burden of unfolded proteins in the ER (ER stress) have long been suspected to exist (*Freiden et al., 1992*; *Gaut, 1997*; *Laitusis et al., 1999*), but the details have only come into focus recently.

BiP binding to its clients is in competition with inactivating oligomerization due to self-association via substrate interactions amongst individual BiP molecules. BiP oligomers likely serve as a repository from which pre-existing active chaperone can be rapidly recruited when the concentration of unfolded clients increases (*Preissler et al., 2015a*). A second mechanism involves the covalent modification of BiP by reversible AMPylation (*Ham et al., 2014*; *Sanyal et al., 2015*). When the burden of unfolded proteins in the ER declines, the ER-localized enzyme FICD (or HYPE) uses ATP to transfer adenosine monophosphate (AMP) onto the hydroxyl side chain of a single residue within BiP,

threonine 518 (T518) (*Preissler et al., 2015b*). With mounting ER stress, an altered functional form of the same enzyme, FICD, rapidly removes the AMP, restoring the hydroxyl side chain to T518 and BiP to its ground state (*Preissler et al., 2017*).

Deregulated FICD activity strongly induces the UPR; an observation consistent with modified BiP being less able to buffer ER stress (*Ham et al., 2014*; *Preissler et al., 2015b*; *Preissler et al., 2017*). The inactivating nature of the modification is further supported by observations that in vitro AMPylated BiP binds a model peptide substrate less stably than the unmodified chaperone and by evidence for an ill-defined defect in the responsiveness of AMPylated BiP to the stimulatory effect of J-domain proteins (*Preissler et al., 2015b*).

Like other members of its family, BiP consists of an N-terminal nucleotide binding domain (NBD) and a C-terminal substrate binding domain (SBD), both of which are connected by a conserved hydrophobic linker. BiP, and Hsp70s in general, interact transiently with extended, usually hydrophobic, amino acid sequences exposed on the surface of their client proteins (*Blond-Elguindi et al., 1993*; *Flynn et al., 1991*). These substrate interactions are modulated by nucleotide binding and hydrolysis at the NBD that are accompanied by substantial conformational changes in both domains (*Mayer, 2013*). Substrate binding is facilitated by J-domain containing co-chaperones (J-proteins) that stimulate ATP hydrolysis by Hsp70s in proximity of the substrate. The concerted action of chaperone and J-protein yields a substrate binding machine of ultra-high affinity, characterized by fast initial association of the Hsp70/BiP with the substrate (high 'on' rates) in the ATP-bound state and slow dissociation (low 'off' rates) upon J domain-driven ATP hydrolysis (*De Los Rios and Barducci, 2014*; *Misselwitz et al., 1998*).

Here we have combined biochemical and structural approaches to dissect how AMPylation influences the J protein-driven ATPase cycle of BiP to cause its functional inactivation. Our findings reveal an unanticipated role for the structural modification of a residue on the far reaches of BiP's SBD - the AMPylated T518 - in affecting an allosteric transition that is essential for Hsp70s to achieve ultra-affinity for binding their substrates.

## Results

### AMPylation biases BiP towards a conformation normally attained by the ATP-bound chaperone

Nucleotide binding and hydrolysis markedly affect interactions between the NBD and SBD of Hsp70s/BiP. In both the ADP-bound and nucleotide-free (or apo) state the two domains are undocked, substrates are bound tightly (with low 'off' rates), and the hydrophobic interdomain linker is exposed to the solvent (*Bertelsen et al., 2009*). In the ATP-bound state the two domains are docked against each other, substrates exchange rapidly (with high 'on' and 'off' rates), and the interdomain linker is tucked against the NBD and relatively shielded from the solvent (*Kityk et al., 2012*; *Kumar et al., 2011*; *Qi et al., 2013*; *Yang et al., 2015*). These mechanisms initially characterized in the bacterial DnaK likely extend to all members of the family, including BiP.

The bacterial protease SubA has remarkable specificity for BiP's interdomain linker, cleaving it between L416 and L417 (*Paton et al., 2006*). It is thus a useful tool to probe the linker's disposition vis-à-vis the aforementioned allosteric transitions. However, the tendency of ADP-bound (or apo) BiP to oligomerize confounds interpretation of the effects of nucleotide on SubA-mediated linker cleavage, because engagement of the interdomain linker of one BiP molecule in the SBD of another protects the linker from cleavage (*Preissler et al., 2015a*). The net result of these two competing processes manifests in complete resistance of a substantial fraction of ADP-bound wildtype BiP (BiP[WT]) to cleavage by SubA (*Preissler et al., 2015a*', and reaction 1 in *Figure 1* and *Figure 1—figure supplement 1*). Interestingly, AMPylation rendered the otherwise refractory fraction of ADP-bound BiP susceptible to cleavage (compare reactions 1 and 5 in *Figure 1*, and *Figure 1—figure supplements 1*, *2* and *3*).

The competing effect of linker protection by inter-molecular engagement in the SBD of ADP-bound BiP oligomers can be nearly eliminated by a mutation in the SBD, V461F, that enfeebles substrate binding without affecting the allosteric transitions caused by nucleotide binding and hydrolysis (*Preissler et al., 2015a*). When the same experiment was performed with an ADP-bound substrate binding-deficient BiP[V461F] mutant, the linker was rapidly cleaved nearly to completion. Cleavage of

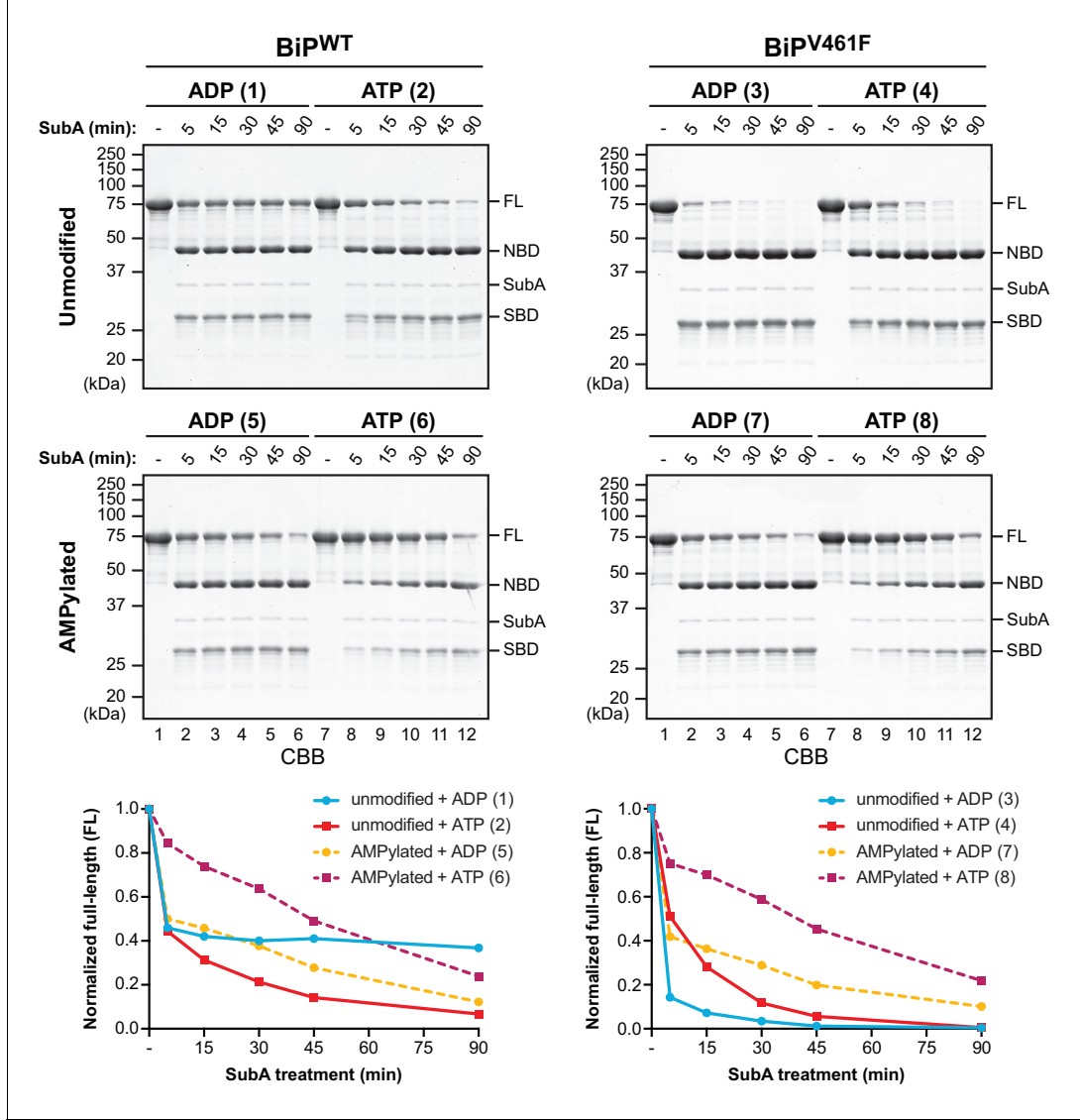

**Figure 1.** AMPylation biases BiP towards an ATP bound-like conformation. Coomassie-stained (CBB) SDS-PAGE gels of purified unmodified or AMPylated wildtype BiP (BiP$^{WT}$) and oligomerization-deficient V461F mutant (BiP$^{V461F}$) proteins digested with the linker-specific SubA protease for the indicated times in presence of ADP or ATP. The intact proteins (FL) and isolated nucleotide binding domain (NBD) and substrate binding domain (SBD) are indicated. The time-dependent change of the intact BiP (FL) signals were quantified, normalized to the initial value (set to 1), and plotted in the graphs below. The numbering (1–8, in parentheses) refers to the explanatory cartoons of these limited proteolysis reactions that analyze the nucleotide-dependent conformational states of unmodified and AMPylated BiP (*Figure 1—figure supplement 1*). Shown are representative experimental observations reproduced three times (*Figure 1—figure supplement 2*).

DOI: https://doi.org/10.7554/eLife.29428.002

The following figure supplements are available for figure 1:

**Figure supplement 1.** Cartoons depicting the conformational states of BiP in the reactions presented in *Figure 1*.
DOI: https://doi.org/10.7554/eLife.29428.003

**Figure supplement 2.** AMPylation alters the sensitivity of BiP to cleavage by SubA.
DOI: https://doi.org/10.7554/eLife.29428.004

**Figure supplement 3.** Analysis of unmodified and AMPylated BiP by mass spectrometry.
DOI: https://doi.org/10.7554/eLife.29428.005

BiP[V461F] was slower in the presence of ATP, consistent with a domain-docked conformation that partially protects the linker from cleavage (compare reactions 3 and 4 in *Figure 1* and *Figure 1—figure supplements 1* and *2*). AMPylation, which exposed the interdomain linker of ADP-bound BiP[WT] to cleavage by SubA, had a stabilizing effect on the interdomain linker of the ADP-bound mutant BiP[V461F] (compare reactions 3 and 7 in *Figure 1* and *Figure 1—figure supplements 1* and *2*), resulting in very similar cleavage kinetics of AMPylated BiP[WT] and AMPylated BiP[V461F] in both nucleotide states (compare the lower panels of *Figure 1*).

These observations are consistent with the notion that AMPylation biases BiP towards a domain-docked conformation with higher substrate exchange kinetics (the substrate being the interdomain linker in these reactions, set up in the absence of other clients). Thus, in the case of ADP-bound BiP[WT], AMPylation likely impaired substrate interaction-dependent oligomerization and thereby increased BiP's cleavability by SubA, whereas the protective effect of AMPylation on the interdomain linker due to enhanced domain docking dominated in the context of the V461F mutation. The enhanced protection of the linker of ATP-bound AMPylated BiP[WT] and AMPylated BiP[V461F] (compared to their non-AMPylated ATP-bound versions) was a conspicuous and reproducible finding (compare reactions 6 and 8 with 2 and 4 in *Figure 1* and *Figure 1—figure supplements 1* and *2*). These observations suggest that AMPylation not only influences the ADP-bound state to resemble the ATP-bound state but also accentuates features normally found in the ATP-bound chaperone.

## Structure of AMPylated BiP

The limited proteolysis experiments suggested that further insight into the consequences of AMPylation on BiP's conformation might be provided by a structure of the AMPylated chaperone in the apo or ADP-bound state, in which AMPylation exerts its most prominent effect. Most crystal structures of intact Hsp70 chaperones have been obtained in the ATP state, but a previous success in crystallizing an ADP-P$_i$-bound bacterial Hsp70 (*Geobacillus kaustophilus* DnaK) in the domain-undocked conformation has been reported (*Chang et al., 2008*). To facilitate crystallization we deleted the N-terminal nine unstructured residues from mature Chinese hamster BiP as well as a large part of the flexible helical lid (SBDα) at the C-terminus (*Figure 2A*), a deletion that favored crystallization of the ADP-bound *G. kaustophilus* DnaK. Truncation of the lid enhances the substrate binding and release kinetics of DnaK and BiP but preserves nucleotide-dependent allosteric regulation (*Buczynski et al., 2001*; *Chambers et al., 2012*; *Misselwitz et al., 1998*; *Pellecchia et al., 2000*). Additionally, we introduced a T229A mutation that strongly inhibits the ATPase activity of BiP (*Gaut and Hendershot, 1993*; *Wei et al., 1995*) and favors FICD-mediated AMPylation over ATP hydrolysis (*Preissler et al., 2015b*). The bacterially expressed and affinity-purified protein was AMPylated by exposure to active FICD[E234G] in presence of ATP, followed by addition of EDTA to remove the bound nucleotide and size-exclusion chromatography to separate BiP from FICD[E234G]. Stoichiometric AMPylation of BiP was confirmed by intact protein mass spectrometry (see Materials and methods section).

Crystals of AMPylated BiP diffracted well and a structural model of 1.9 Å resolution was derived (PDB 5O4P; *Table 1*). In the crystals BiP adopted a domain-docked conformation. Accordingly, the interdomain linker was well structured and bound to the NBD by parallel β-augmentation (*Figure 2B*). The main-chain conformation of AMPylated BiP was nearly identical to that of ATP-bound full-length human BiP (PDB 5E84) (*Yang et al., 2015*) (Cα alignment RMS = 0.551 Å; *Figure 2C*). However, the nucleotide-binding pocket in the NBD of AMPylated BiP was occupied by several water molecules and a sulfate group was found at the position where the γ-phosphate of ATP would otherwise be located (*Figure 2D* and *Figure 2—figure supplement 1A*).

Although the crystalized AMPylated BiP was clearly ATP-free, it adopted a conformation that is characteristic of the ATP-bound state. This surprising feature extends to the fine details, as the NBD of nucleotide-free AMPylated BiP assumed a conformation that resembled the ATP-bound NBD of intact BiP (*Yang et al., 2015*) (*Figure 2C and D*) and deviated considerably in structure from the isolated NBD, whether crystallized in the apo state or with ADP or ATP (*Macias et al., 2011*; *Wisniewska et al., 2010*) (*Figure 2—figure supplement 1B*). This feature is likely imparted on the NBD allosterically, as a consequence of AMPylation, because the T229A mutation did not abolish the nucleotide-dependent allosteric regulation of BiP nor bias the protein towards an ATP-bound conformation (*Figure 2—figure supplement 2A and B*).

As expected of BiP/Hsp70 in the domain-docked conformation, the truncated lid of the SBD (SBDα) was in the open position and the substrate binding groove was substantially wider than that

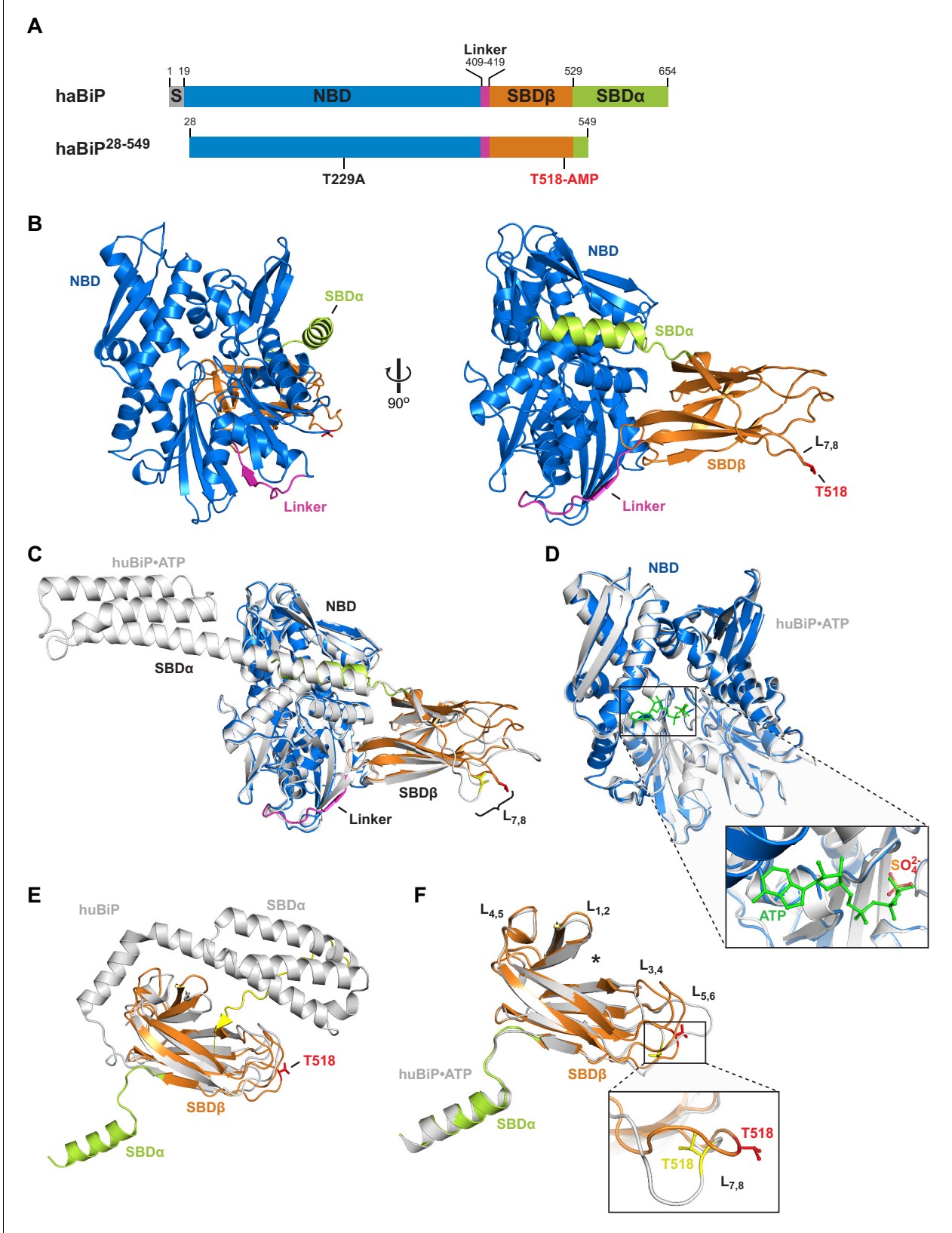

**Figure 2.** Crystal structure of in vitro AMPylated apo BiP. (**A**) Schematic representation of Chinese hamster BiP (haBiP) and the derivative used for crystallization (haBiP²⁸⁻⁵⁴⁹). The following features are indicated: signal sequence (grey; S), nucleotide binding domain (blue; NBD), interdomain linker

*Figure 2 continued on next page*

*Figure 2 continued*

(magenta; Linker), substrate binding subdomain β (orange; SBDβ), substrate binding subdomain α (green; SBDα), and the AMPylation site (T518-AMP). (**B**) Ribbon representation of the structure of crystallized haBiP[28-549] (PDB 5O4P) in two orientations with coloring as in 'A'. (**C**) Comparison of haBiP[28-549] (PDB 5O4P) with the structure of unmodified ATP-bound human BiP (huBiP, PDB 5E84; grey). Loop $L_{7,8}$ comprising the AMPylation site (T518) is indicated, otherwise coloring as in 'A'. (**D**) Same as in 'C' showing only the NBDs. The inset is a close-up view of the nucleotide binding cleft. The ATP (green) bound by the unmodified huBiP and a sulfate bound by AMPylated haBiP (blue) are indicated in stick diagram. Note that the sulfate group fully occupies the position of the terminal γ-phosphate of the bound ATP. (**E**) Comparison of the SBD of AMPylated haBiP[28-549] (PDB 5O4P; coloring as in 'A') with the isolated SBD of unmodified huBiP (PDB 5E85, grey). A C-terminal substrate peptide (yellow) occupies the peptide binding groove of unmodified huBiP. The AMPylation site (T518) is indicated (red). (**F**) As in 'C' comparing only the substrate binding domains. The asterisk (*) marks the peptide binding groove. The inset shows a close-up view of $L_{7,8}$. Note the outward orientation of the T518 side chain in the AMPylated haBiP[28-549] (red) compared to its inward orientation in unmodified huBiP (yellow).

DOI: https://doi.org/10.7554/eLife.29428.006

The following figure supplements are available for figure 2:

**Figure supplement 1.** Features of the AMPylated apo BiP structure point to the absence of nucleotide in the NBD and AMPylation of T518.

DOI: https://doi.org/10.7554/eLife.29428.007

**Figure supplement 2.** The T229A mutation does not alter BiP's nucleotide-dependent allostery.

DOI: https://doi.org/10.7554/eLife.29428.008

**Figure supplement 3.** Structures of AMPylated BiP in a different crystal form and in the presence of diverse ligands exhibit consistent modification-dependent features.

DOI: https://doi.org/10.7554/eLife.29428.009

**Figure supplement 4.** Mass spectrometry analysis of BiP used for crystallization.

DOI: https://doi.org/10.7554/eLife.29428.010

**Figure supplement 5.** Crystal structure of ADP-bound AMPylated BiP.

DOI: https://doi.org/10.7554/eLife.29428.011

of the isolated SBD of human BiP (PDB 5E85, which reflects the state of the SBD of the domain-undocked ADP-bound chaperone, *Bertelsen et al., 2009*; *Yang et al., 2015*) (*Figure 2E*). Differences between the SBD of AMPylated apo BiP and ATP-bound BiP were also noted in the distal loops of the SBDβ subdomain (*Figure 2E and F*). In particular the conformation of loop $L_{7,8}$, harboring the AMPylation site residue T518, differed substantially between both structures. Although no clear density for the entire AMP moiety was observed, several indications point to the presence of the modification. First, the side chain of T518 is oriented outwards and protrudes into the solvent (*Figure 2F*). The missing density may thus be explained by flexibility of the hydrophilic AMP exposed on the protein surface. Second, additional electron density extends from the T518 side chain hydroxyl group, consistent with the phosphodiester linkage to adenosine (*Figure 2—figure supplement 1C*). Third, whereas the surrounding loops are engaged with the neighboring molecule at the crystal packing interface, loop $L_{7,8}$ does not form such contacts, providing space in the crystal to accommodate AMP and affording substantial flexibility to the region, as reflected in the high B-factors of $L_{7,8}$ of AMPylated BiP (*Figure 2—figure supplement 1D*).

The aforementioned features were observed in four additional crystal forms of AMPylated BiP derived from independent protein preparations and from molecules arranged in a different space group (*Table 1* and *Figure 2—figure supplement 3A*) and supported by intact protein mass spectrometry, pointing to the presence of the modification in the crystallized protein (*Figure 2—figure supplement 4A and B*). In all structures BiP adopted a very similar domain-docked conformation in absence of density for ATP in the NBD and all reveal additional density protruding from the T518 side chain (*Figure 2—figure supplement 3A–C*). Furthermore, their $L_{7,8}$ loop regions appeared more disordered than surrounding parts of the SBDβ (indicated by higher B-factors, *Figure 2—figure supplement 3D*). These findings indicate that the bulky modification favored solvent exposure of loop $L_{7,8}$ and prevented T518 from forming intramolecular contacts observed in structures of unmodified BiP or other Hsp70s.

Notably, a structure of AMPylated BiP from a crystallization reaction supplemented with ADP showed that BiP adopted the domain-docked, loop $L_{7,8}$-exposed conformation even when ADP was bound (*Table 1* and *Figure 2—figure supplement 5A–E*). Thus, despite lack of density corresponding to the bulk of the modification, the structural data are in agreement with the results of the limited proteolysis experiments (*Figure 1*) and strongly support the conclusion that AMPylation favors a

**Table 1.** Data collection and refinement statistics

| AMPylated haBiP[28-549] | Apo | Apo | Apo | ADP | ADP |
|---|---|---|---|---|---|
| Data collection | | | | | |
| Synchrotron stations (DLS) | I02 | I24 | I24 | I24 | I24 |
| Space group | $P2_1$ | $P2_12_12_1$ | $P2_12_12_1$ | $P2_12_12_1$ | $P2_12_12_1$ |
| Cell dimensions | | | | | |
| a,b,c; (Å) | 68.33, 118.65, 83.15 | 64.65, 69.07, 122.23 | 69.088, 75.408, 98.308 | 69.089, 75.310, 98.350 | 69.05, 75.61, 97.50 |
| $\alpha, \beta, \gamma$; (°) | 90, 97, 90 | 90, 90, 90 | 90, 90, 90 | 90, 90, 90 | 90, 90, 90 |
| Resolution, (Å) | 67.76–1.86 (1.91–1.86)* | 61.12–2.0 (2.05–2.00) | 75.41–1.67 (1.7–1.67) | 98.35–1.71 (1.74–1.71) | 97.5–1.59 (1.61–1.59) |
| $R_{merge}$ | 0.089 (0.83) | 0.158 (1.31) | 0.086 (0.794) | 0.096 (1.002) | 0.131 (0.906) |
| $R_{meas}$ | 0.122 (1.119) | 0.185 (1.532) | 0.096 (0.871) | 0.108 (1.104) | 0.145 (1.022) |
| $<I/\sigma (I)>$ | 9.6 (1.4) | 8.5 (1.5) | 12 (2.1) | 13 (2.2) | 8.8 (2.2) |
| $CC_{1/2}$ | 0.995 (0.533) | 0.995 (0.522) | 0.997 (0.888) | 0.998 (0.820) | 0.990 (0.836) |
| Completeness, % | 99.4 (99.1) | 99.9 (100) | 98.3 (100) | 99.9 (100) | 100 (99.9) |
| Redundancy | 3.4 (3.5) | 7.1 (7.2) | 6.6 (6.7) | 6.6 (6.7) | 6.5 (6.4) |
| Refinement | | | | | |
| $R_{work}$ | 0.195 | 0.240 | 0.210 | 0.210 | 0.200 |
| $R_{free}$ | 0.212 | 0.280 | 0.250 | 0.250 | 0.220 |
| No. of reflections | 104104 | 35827 | 56476 | 53637 | 66447 |
| No. of atoms | 8750 | 3994 | 4281 | 4245 | 4156 |
| Average B-factors | 24.9 | 35.7 | 23.14 | 23.3 | 30.09 |
| RMS deviations | | | | | |
| Bond lengths (Å) | 0.007 | 0.007 | 0.007 | 0.009 | 0.008 |
| Bond angles, (°) | 1.261 | 1.151 | 1.184 | 1.485 | 1.332 |
| Ramachandran favoured region, % | 99 | 97.49 | 98.05 | 98.44 | 99.03 |
| Ramachandran outliers, % | 0.19 | 0 | 0 | 0 | 0 |
| MolProbity score (percentile[†]) | 0.67 (100%) | 0.83 (100%) | 0.67 (100%) | 0.78 (100%) | 0.74 (100%) |
| PDB code | 5O4P | 6EOB | 6EOC | 6EOE | 6EOF |

* Values in parentheses are for the highest-resolution shell.

[†] 100th percentile is the best among structures of comparable resolution.

DOI: https://doi.org/10.7554/eLife.29428.012

state of BiP that, in absence of ATP, is very similar to the conformation of ATP-bound unmodified BiP.

## AMPylation impairs BiP oligomerization

The data presented above reveal that AMPylation biases BiP's conformational equilibrium towards an ATP bound-like, domain-docked state that is predicted to weaken substrate binding. BiP molecules bind each other through typical chaperone-substrate interactions. In cells, these interactions assemble inactive BiP oligomers to buffer short-term fluctuations in the ER unfolded protein load, whereas in vitro BiP oligomerization is a convenient means to probe its affinity for substrates (*Preissler et al., 2015a*). Purified AMPylated BiP migrates mostly as a monomer on native-PAGE gels and AMPylation increases the monomeric fraction of endogenous BiP from mammalian cells, suggesting that the modification might interfere with the ability of BiP to oligomerize (*Preissler et al., 2015b*). To explore this in further detail, we monitored the oligomeric state of BiP by size-exclusion chromatography.

Thereby, monomers of purified BiP were separated from earlier eluting oligomers, which are stabilized by absence of nucleotide or by presence of ADP - with dimers dominating at the concentration tested. In contrast, AMPylated BiP eluted mainly as a monomer (*Figure 3A and B*) consistent

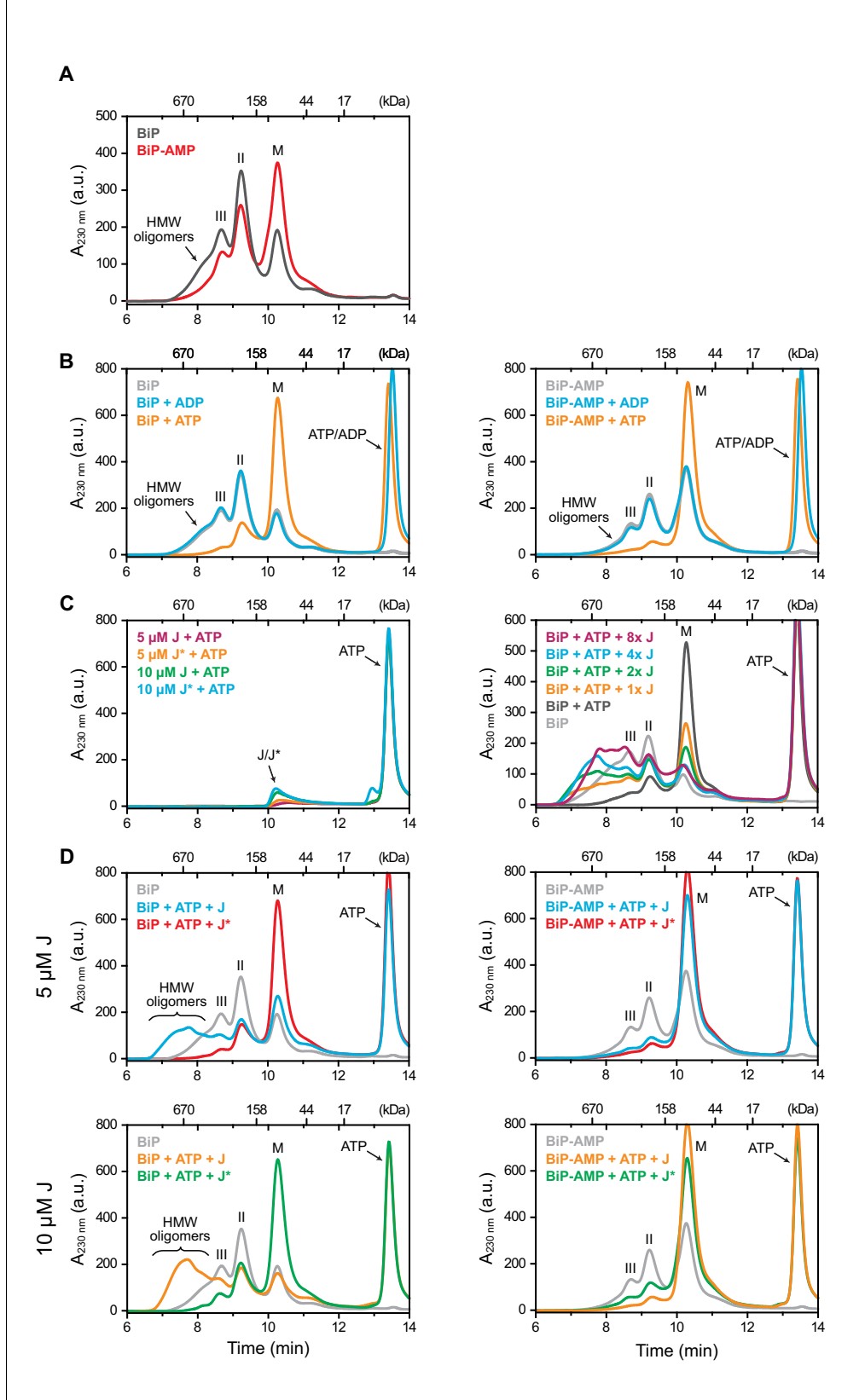

**Figure 3.** AMPylation inhibits J protein-mediated BiP oligomerization. (**A**) Peptide bond absorbance traces ($A_{230\ nm}$) of wildtype unmodified (BiP) and AMPylated (BiP-AMP) BiP proteins (both at 50 μM) resolved by size-exclusion chromatography in absence of added nucleotide. The elution peak

*Figure 3 continued on next page*

Figure 3 continued

corresponding to BiP monomers (M) and earlier eluting peaks (II and III) representing BiP dimers and trimers, respectively, as well as high-molecular weight (HMW) oligomers are indicated. Note the enhanced monomer signal and paucity of oligomers in the sample of AMPylated BiP. (B) As in 'A' but in presence of ADP or ATP (both at 1.5 mM). Note the nearly complete disassembly of BiP oligomers in the ATP-containing sample. (C) Where indicated the same experiment as in 'A' was performed in presence of ATP (1.5 mM) and different concentrations of the wildtype (J) or inactive QPD mutant (J*) J-domain of ERdj6 (1 × represents 1.25 µM of J-domain). Note the J-domain concentration-dependent increase in peaks II and III and the formation of faster eluting species (before 8 min) representing large BiP oligomers. (D) Comparison of the elution profiles of unmodified (left) and AMPylated BiP (right) treated as in 'C'. Note the near absence of early-eluting oligomeric species in the AMPylated BiP sample. A representative experiment of three independent repeats is shown.

DOI: https://doi.org/10.7554/eLife.29428.013

with previous evidence of impaired substrate interactions (*Preissler et al., 2015b*). Despite its bias towards the conformation normally imposed by ATP binding, AMPylated BiP retained further responsiveness to ATP, as the remaining oligomers of both unmodified and modified BiP disassembled into monomers upon exposure to ATP (*Figure 3B*).

Basal ATP hydrolysis rates of Hsp70s are low. Under physiological conditions this process requires stimulation by co-chaperones, which share a structurally conserved J-domain that interacts with the chaperone to promote ATP hydrolysis and thereby stable substrate binding (*Liberek et al., 1995*). In the absence of other substrates, addition of a J-domain promotes oligomerization of Hsp70, as J-stimulated ATP hydrolysis (and the relatively slow intrinsic exchange of ADP for ATP) dynamically enforces a regime of enhanced substrate interactions between individual chaperone molecules (*King et al., 1995*; *King et al., 1999*). This feature was therefore exploited to investigate the functional consequences of BiP AMPylation on J-mediated substrate binding.

In the presence of ATP, increasing concentrations of the J-domain of ERdj6 purified as a glutathione S transferase (GST) fusion protein (GST-J$^{WT}$, *Petrova et al., 2008*) progressively shifted the equilibrium from monomers towards BiP oligomers (*Figure 3C*). J-mediated oligomers were much larger than those observed in the absence of nucleotide or in presence of ADP (compare *Figure 3C* with *Figure 3A and B*). BiP oligomerization was dependent on the functionality of the J-domain, as no oligomers were established by a mutant J-domain in which the histidine residue of its conserved HPD motif - essential for stimulation of the ATPase activity of Hsp70s (*Wall et al., 1994*) - was exchanged to glutamine (GST-J$^{QPD}$; *Figure 3D*). Importantly, J domain-mediated BiP oligomerization was nearly absent when AMPylated BiP was tested in the assay (right panels in *Figure 3D*). Thus, AMPylation interfered with BiP's cooperation with a J-domain co-chaperone to form a stable substrate interaction.

## AMPylation of BiP affects its J protein-mediated interactions with substrates

BiP oligomerization represents a particular type of substrate interactions; to study J domain-mediated substrate interactions by an alternative approach we took advantage of the well-documented ability of Hsp70s (*Mayer et al., 1999*; *Suh et al., 1998*) and BiP in particular to recognize J-proteins as substrates in vitro (*Misselwitz et al., 1999*). Accordingly, in the presence of ATP, BiP$^{WT}$ co-purified efficiently with GST-J$^{WT}$ coupled to glutathione sepharose beads in a co-purification assay (*Petrova et al., 2008*', and lane 8 in *Figure 4A*). This interaction was observed neither in presence of ADP nor when mutant GST-J$^{QPD}$ was immobilized, nor when substrate binding-deficient BiP$^{V461F}$ was used (*Figure 4A*). The assay therefore reported on a J protein-dependent substrate interaction of BiP. Importantly, AMPylated BiP did not bind stably to beads carrying GST-J$^{WT}$ in presence of ATP, pointing to defective J protein-mediated high-affinity substrate interactions imposed by the modification (lane 10 in *Figure 4A*).

Consistent with its bias towards the ATP-like state, AMPylated BiP retains the ability to bind substrates, but with accelerated release kinetics (*Preissler et al., 2015b*', Figure 8E therein). To circumvent the limitation of post-equilibrium methods (such as co-purification or size-exclusion chromatography) to report on transient substrate interactions, a bio-layer interferometry (BLI) assay was established to monitor J protein-mediated BiP engagement of substrates in real-time. A biotinylated GST-J fusion protein and a model BiP substrate peptide (P15, *Misselwitz et al., 1998*) were co-immobilized on the surface of streptavidin-coated BLI sensors (*Figure 4B*). The sensors were then

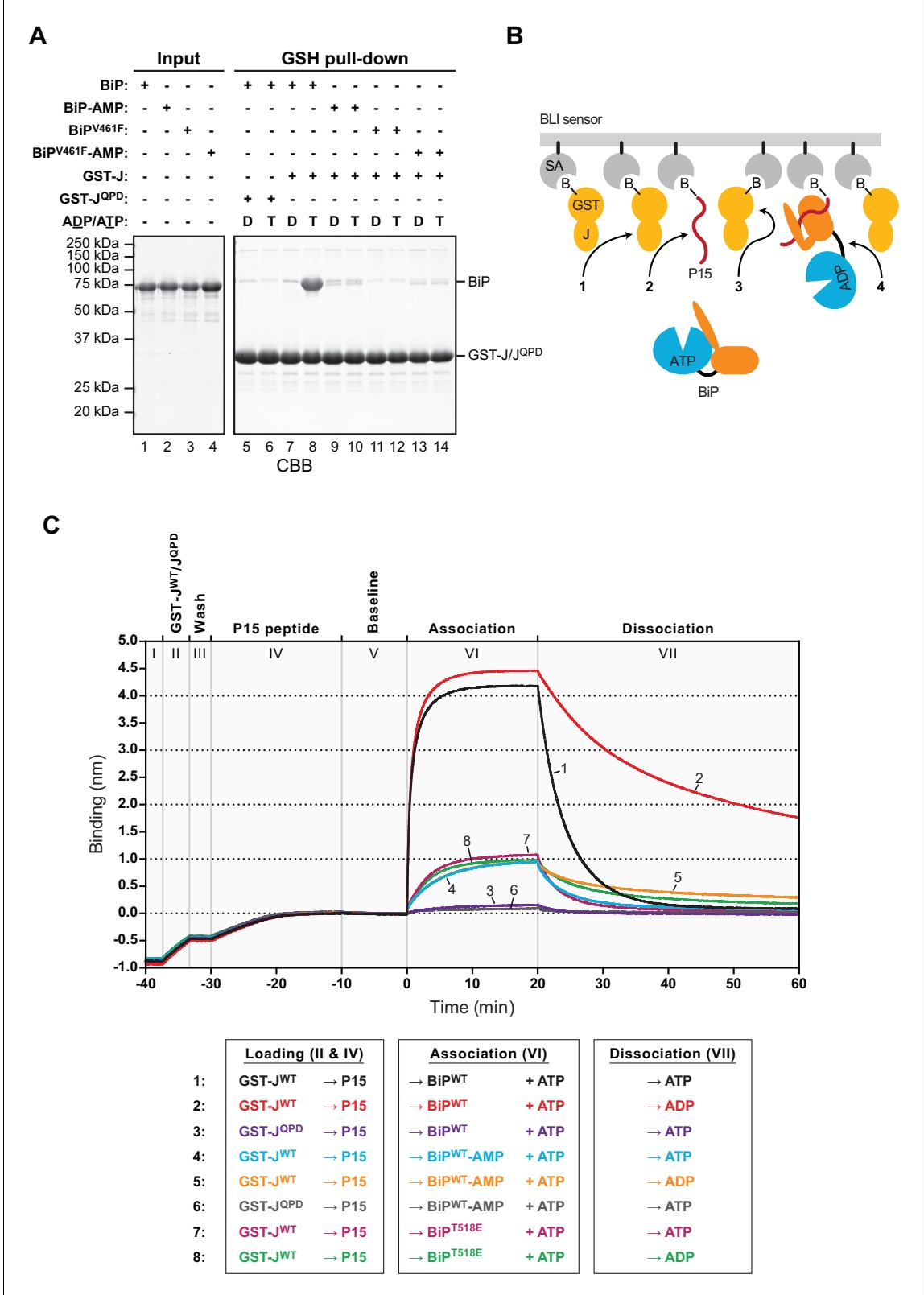

**Figure 4.** AMPylation inhibits J protein-mediated substrate binding. (**A**) Coomassie-stained SDS-PAGE gels of proteins eluted from a glutathione (GSH) sepharose matrix. Unmodified or AMPylated wildtype or V461F mutant BiP proteins were incubated in presence of ADP or ATP with wildtype or QPD mutant J-domain coupled via a GST-fusion protein to GSH sepharose beads and subsequently washed extensively. The input and the bound proteins (GSH pull-down) eluted by denaturation in SDS are indicated. (**B**) Cartoon depicting possible interactions between soluble BiP (the

Figure 4 continued

analyte) and biotinylated GST-J co-immobilized with a BiP substrate peptide (P15) on a bio-layer interferometry (BLI) sensor (the ligand). Streptavidin (SA)-coated BLI sensors (grey), the biotin moiety (B) covalently attached to GST-J (yellow), biotinylated P15 substrate peptide (red), and ATP-bound BiP in solution are shown. Arrows indicate J domain-mediated BiP interactions with GST-J in trans (1), P15 (2), GST-J in cis (3) and J domain-mediated BiP oligomerization (4). (C) Plot of the BLI signal as a function of time in experiments as cartooned in 'B' and tabulated below. The individual steps of the experiment (I-VII) are indicated: After an initial equilibration step (I) biotinylated wildtype or QPD mutant GST-J corresponding to an interference signal difference of ~0.4 nm was immobilized (II). Following a short wash step (III) the binding sites on the sensors were saturated with biotinylated P15 peptide (IV), followed by an extended wash step to establish a stable baseline signal (V). The sensors were then introduced into solutions containing unmodified (BiP) or AMPylated (BiP-AMP) wildtype BiP or BiP$^{T518E}$ mutant proteins to detect their association with the sensors in presence of ATP (VI). Dissociation of BiP from the sensor was measured in protein-free solutions containing either ADP or ATP (VII). A representative experiment is shown and the same result was observed in at least three independent repeats. Note that in the presence of ADP the dissociation of AMPylated BiP ($32.9 \times 10^{-4} \pm 19.0 \times 10^{-4}$ s$^{-1}$; trace 5) and BiP$^{T518E}$ ($16.5 \times 10^{-4} \pm 7.3 \times 10^{-4}$ s$^{-1}$, trace 8) was faster than unmodified BiP ($7.9 \times 10^{-4} \pm 5.3 \times 10^{-4}$ s$^{-1}$; trace 2). The dissociation rate constants represent mean values with standard deviations.

DOI: https://doi.org/10.7554/eLife.29428.014

The following figure supplement is available for figure 4:

**Figure supplement 1.** J domain-mediated substrate interactions depend on BiP's ATPase activity and ability to bind substrates.

DOI: https://doi.org/10.7554/eLife.29428.015

introduced into solutions containing BiP to detect its association, followed by transfer to BiP-free solutions to record dissociation. In presence of ATP unmodified BiP$^{WT}$ was rapidly recruited to sensors carrying GST-J$^{WT}$ and P15 (traces 1 and 2 in *Figure 4C*). This interaction required a functional J-domain (trace 3 in *Figure 4C*) and was dependent on BiP's ability both to bind substrate and to hydrolyze ATP (*Figure 4—figure supplement 1*).

Dissociation of the bound BiP$^{WT}$ was faster in presence of ATP compared with ADP (*Figure 4C*, compare dissociation phase of traces 1 and 2). This finding indicated that the apparent plateau in binding (observed in the preceding association phase, with both BiP and ATP present in solution) was enforced by cycles of nucleotide exchange-driven BiP dissociation and J protein-driven ATP hydrolysis-dependent rebinding, as formalized in the ultra-affinity model for J-mediated Hsp70-substrate interactions (*De Los Rios and Barducci, 2014*). In contrast, AMPylated BiP$^{WT}$ showed a severely reduced binding plateau to GST-J$^{WT}$-coupled sensors in presence of ATP and its dissociation kinetics were faster in presence of ADP compared to unmodified BiP$^{WT}$ (traces 4 and 5 in *Figure 4C*).

Mutant BiP$^{T518E}$, which mimics aspects of AMPylation (*Preissler et al., 2015b*), behaved similarly to AMPylated BiP$^{WT}$ in this assay (compare traces 7 and 8 to 4 and 5 in *Figure 4C*). ATP further accelerated dissociation of the bound AMPylated BiP$^{WT}$ and BiP$^{T518E}$, paralleling unmodified BiP$^{WT}$. These BLI experiments and the oligomerization assays both reveal a consistent defect in the ability of modified BiP to achieve J protein-mediated ultra-affinity substrate binding, whilst showcasing the residual responsiveness of the modified protein to ATP.

## AMPylation of BiP does not alter its non-substrate interactions with the J-domain

The defect in J protein-mediated substrate interaction imposed by AMPylation may have arisen from impairment in the initial interaction between AMPylated BiP and the J-domain or from a defect in subsequent stimulated ATP hydrolysis (the functional output of the initial engagement), or a combination of both. To distinguish amongst these possibilities we needed to deconvolute the two components that contribute to the binding of BiP to immobilized J-protein: (i) the initial engagement of the J-domain with ATP-bound BiP (involving non-substrate, protein-protein interactions) and (ii) typical substrate interactions catalyzed by J protein-mediated ATP hydrolysis; the latter is expected to dominate in binding assays (*Mayer et al., 1999*; *Suh et al., 1998*).

In an effort to measure the non-substrate interaction of BiP with the J-domain the T229A mutation (that inhibits ATP hydrolysis without affecting nucleotide binding-coupled allostery, *Figure 2—figure supplement 2*) and the V461F mutation (that inhibits substrate binding without affecting allostery, *Figure 1* and *Figure 2—figure supplement 2*) were combined to generate BiP$^{T229A-V461F}$. BLI sensors coated with immobilized GST-J$^{WT}$ (in absence of a substrate peptide) were exposed to different concentrations of unmodified or AMPylated BiP$^{T229A-V461F}$ in presence of ATP. Plotting the

steady state binding amplitudes against the BiP concentrations revealed very similar binding constants ($K_d \approx$ 14 µM; *Figure 5*). The similar shape of the curves also indicated that AMPylation did not significantly alter the association or dissociation kinetics.

Several control experiments confirmed that substrate interactions had indeed a negligible contribution to the observed binding of BiP$^{T229A-V461F}$ to GST-J$^{WT}$: First, the dissociation rate of the

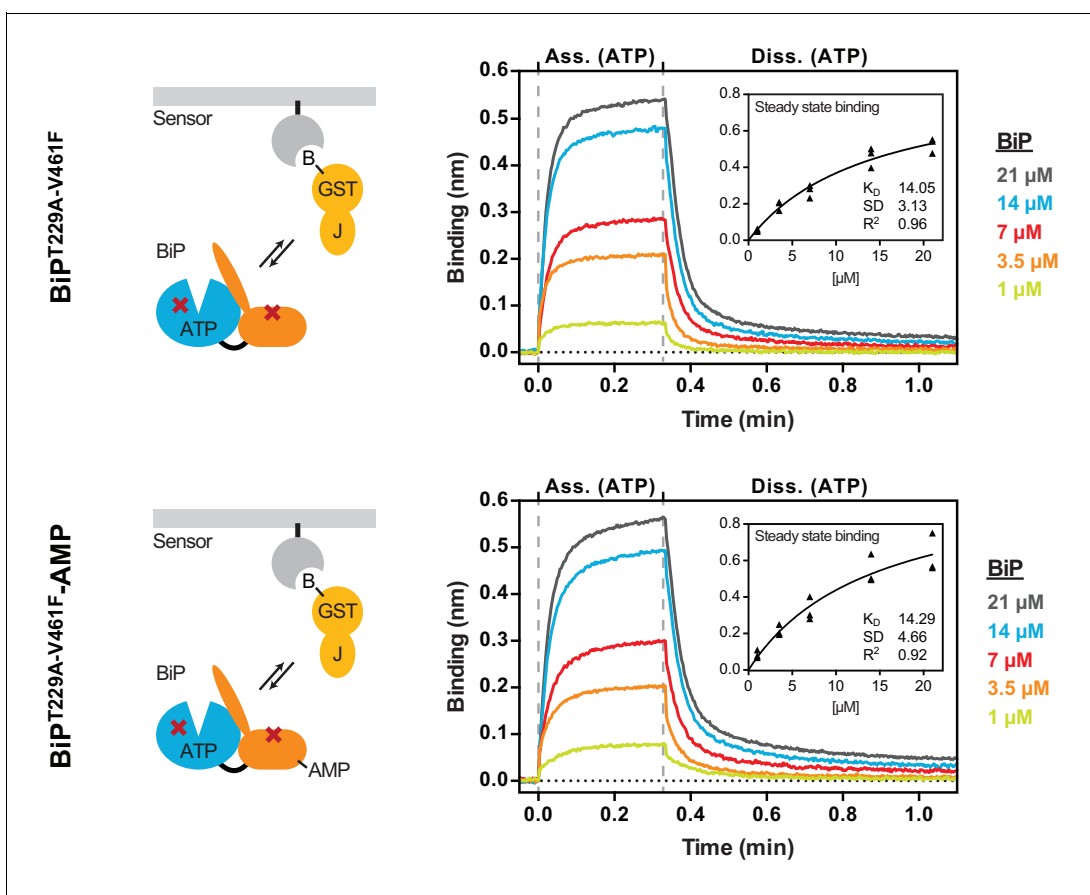

**Figure 5.** AMPylation does not alter non-substrate interactions between BiP and a model J-domain. Bio-layer interferometry (BLI) experiment based on the experiment outlined in *Figure 4C* to measure the functionally relevant non-substrate interactions between an ATPase and substrate binding BiP mutant protein (BiP$^{T229A-V461F}$) and a model J-domain fused to GST (GST-J). The cartoons on the left depict the anticipated interactions between BiP in solution and the J-domain on the sensor surface during the association step and plots of BLI binding signals against time are shown on the right. After an initial equilibration step the sensors were saturated with biotinylated GST-J followed by another wash step to achieve a stable baseline signal (not shown). The sensors were then exposed to solutions containing different concentrations of unmodified or AMPylated BiP$^{T229A-V461F}$ proteins to measure their association with the sensors. Dissociation was detected in protein-free solutions containing ATP. Representative plots of recorded binding signals of the association and dissociation steps are shown. The insets present plots of the plateau binding amplitudes during the association step against BiP concentrations of three independent experiments as well as the obtained dissociation constant ($K_D$) values with the corresponding standard deviations and the R$^2$ values of the fits.

DOI: https://doi.org/10.7554/eLife.29428.016

The following figure supplements are available for figure 5:

**Figure supplement 1.** The non-substrate interaction between BiP$^{T229A-V461F}$ and the BLI sensor requires a functional J-domain.

DOI: https://doi.org/10.7554/eLife.29428.017

**Figure supplement 2.** The non-substrate interaction between BiP$^{T229A-V461F}$ and the J-domain is sensitive to elevated salt concentrations.

DOI: https://doi.org/10.7554/eLife.29428.018

complex was very similar in presence of ADP and ATP (*Figure 5—figure supplement 1*; a stark contrast to the nucleotide-dependent dissociation kinetics of BiP$^{WT}$ from BLI sensors loaded with GST-J$^{WT}$ and P15, *Figure 4C*). Second, consistent with the predicted ionic character of the interaction between the HPD motif of the J-domain and BiP/Hsp70, binding was sensitive to increasing salt concentrations, whereas substrate interactions between BiP$^{WT}$ and the sensor remained nearly unaffected (*Figure 5—figure supplement 2A and B*). Thus, AMPylation did not alter BiP's HPD motif-dependent, non-substrate interaction with the J-domain.

## AMPylation inhibits the J protein-stimulated ATPase activity of BiP, the rate-limiting step of its nucleotide-driven cycle

The observations described in the previous section suggested that defective J protein-stimulated ATP consumption by AMPylated BiP, observed in a relatively crude, cumulative assay (*Preissler et al., 2015b*', Figure 8C therein), arose from interference with the ATP hydrolysis step that follows normal engagement of the J-domain. To investigate further the direct effect of AMPylation on BiP's J protein-stimulated ATPase activity, the rates of an individual round of ATP hydrolysis were measured in a 'single-turnover' format. In presence of non-functional GST-J$^{QPD}$, unmodified BiP$^{WT}$ showed a very low basal ATP hydrolysis rate (*Figure 6A*), as expected (*Kassenbrock and Kelly, 1989*; *Mayer et al., 2003*; *Wei and Hendershot, 1995*). Basal ATP turnover of AMPylated BiP$^{WT}$ was only slightly (but not significantly) decreased. In contrast, exposure to a wildtype J-domain increased the ATPase rate of unmodified BiP$^{WT}$ more than 4-fold but had almost no effect on the rate of ATP hydrolysis by AMPylated BiP$^{WT}$ (*Figure 6A*). By uncoupling ATP hydrolysis from nucleotide exchange this assay points directly to an AMPylation-induced defect in the ability of ATP-bound BiP to respond to the stimulatory effect of J-protein on ATPase activity. This defect adequately explains much of the previously observed slower J protein-stimulated ATP hydrolysis by AMPylated BiP.

Nonetheless, the effect of AMPylation was not limited to the ability of BiP to respond to J-protein. Release of ADP (and subsequent binding of ATP) re-establishes the low-affinity (high 'off' rate) state and is therefore crucial to BiP's ability to complete its substrate binding cycle. To determine if AMPylation affects the release of ADP (which in the presence of excess ATP is the rate-limiting step for nucleotide exchange, *Mayer et al., 2003*; *Theyssen et al., 1996*) we compared the dissociation rate of a fluorescent ADP derivative (MABA-ADP) from unmodified and AMPylated BiP in a stopped-flow apparatus connected to a fluorometer. MABA-ADP dissociated from unmodified BiP with a rate constant of $9.7 \times 10^{-2} \pm 1.8 \times 10^{-3}$ s$^{-1}$ (within the range reported previously, *Mayer et al., 2003*). AMPylation imposed a modest but reproducible defect on the release of MABA-ADP ($7.2 \times 10^{-2} \pm 7.1 \times 10^{-3}$ s$^{-1}$; *Figure 6B*). This was likely a direct effect of the modification, rather than a consequence of the inability of AMPylated BiP$^{WT}$ to form oligomers in the ADP-bound state, because unmodified oligomerization-deficient BiP$^{V461F}$ had the same MABA-ADP release rate as unmodified BiP$^{WT}$ ($9.4 \times 10^{-2} \pm 3.1 \times 10^{-3}$; *Figure 6B*).

ER calcium concentrations are high and it has been shown that BiP binds ADP with higher affinity at physiological calcium concentrations (*Lamb et al., 2006*). Moreover, nucleotide exchange factors, such as Grp170, enhance ADP release from BiP in the ER (*Behnke et al., 2015*; *Weitzmann et al., 2006*). Accordingly, addition of calcium to the reaction caused a general (~5 fold) decrease in the MABA-ADP release rate (*Figure 6C*). Slightly slower MABA-ADP release imposed by BiP AMPylation was observed under these more physiological conditions too, whereas AMPylation had no significant effect on Grp170-stimulated ADP release (*Figure 6C* and *Figure 6—figure supplement 1*). Furthermore, under the conditions studied here, the J domain-stimulated ATPase rate of unmodified BiP remained slower than even the rate of spontaneous ADP release. While both steps are impaired by AMPylation, the former is more severely affected. This identifies inhibition of the rate-limiting step of BiP's ATPase cycle as the main mechanism of its functional inactivation by AMPylation.

## Discussion

AMPylation is a reversible modification of BiP that correlates inversely with the burden of unfolded proteins in the ER. In the resting ER of secretory cells up to one of every two molecules of the chaperone may be found in the modified state (*Chambers et al., 2012*', Figure 1 therein). Our efforts to understand the consequences of this physiologically-entrained, high-stoichiometry modification of

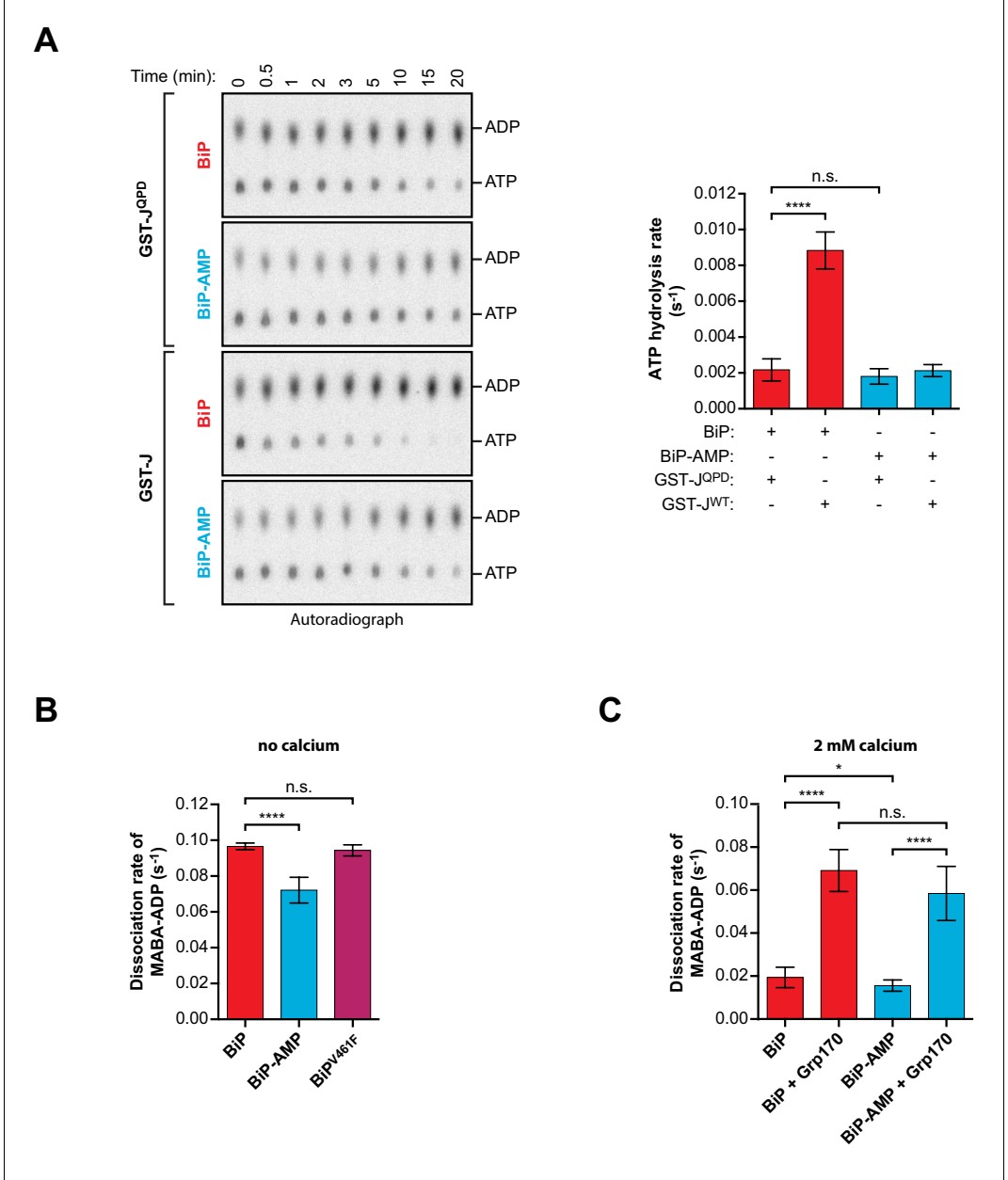

**Figure 6.** Effect of AMPylation on J domain-stimulated ATPase activity of BiP and ADP release from BiP. (**A**) Shown is an autoradiograph of $^{32}$P-labeled ATP and ADP separated by thin-layer chromatography, the products of a single-turnover ATPase assay to analyze the effect of AMPylation on ATP hydrolysis by BiP. Pre-formed complexes between purified unmodified or AMPylated wildtype BiP protein and α-$^{32}$P-ATP were incubated without or with wildtype GST-J or the QPD mutant for the indicated times prior analysis by thin-layer chromatography. A representative experiment is shown on the left and the signals from five repeats of the experiment were quantified and the calculated ATP hydrolysis rates are presented on the graph. Error bars represent standard deviations. ****$p < 0.0001$, n.s. $p > 0.05$. (**B**) Measurement of nucleotide release from BiP in absence of calcium. Unmodified or AMPylated wildtype or V461F mutant BiP proteins were incubated with the fluorescent ADP derivative MABA-ADP and the dissociation of the formed complexes was measured upon dilution with a solution containing excess of ATP to prevent re-binding of MABA-ADP. The dissociation rates of at least three independent repeats are shown. Error bars represent standard deviations. ****$p < 0.0001$, n.s. $p > 0.05$. (**C**) A similar experiment as in 'B' was performed in presence of 2 mM calcium in the solution and without or with Grp170. The dissociation rates of at least five independent repeats are shown. Error bars represent standard deviations. *$p = 0.0281$, ****$p < 0.0001$, n.s. $p > 0.05$.

DOI: https://doi.org/10.7554/eLife.29428.019

The following source data and figure supplement are available for figure 6:

**Source data 1.** Source data and calculated rates for the single-turnover ATPase assays shown in *Figure 6A*.

DOI: https://doi.org/10.7554/eLife.29428.021

*Figure 6 continued on next page*

*Figure 6 continued*

**Source data 2.** Source data and calculated rates for the MABA-ADP release measurements shown in *Figure 6B*.
DOI: https://doi.org/10.7554/eLife.29428.022
**Source data 3.** Source data and calculated rates for the MABA-ADP release measurements shown in *Figure 6C*.
DOI: https://doi.org/10.7554/eLife.29428.023
**Figure supplement 1.** Grp170 stimulates MABA-ADP release from BiP.
DOI: https://doi.org/10.7554/eLife.29428.020

an important ER chaperone have yielded two key findings: Regardless of the identity of the bound nucleotide, AMPylation favors the domain-docked conformation of BiP; a conformation that specifies unstable substrate binding and is normally associated with ATP-bound Hsp70s. In this domain-docked conformation AMPylated BiP is recognized by J-proteins, however, the modification of T518 imposes an intrinsic blockage on BiP's ability to respond to the co-chaperone by accelerated ATP hydrolysis.

## AMPylation weakens BiP-substrate interactions

ADP-bound BiP adopts a heterogeneous ensemble of conformations with a bias towards the domain-undocked state, whereas ATP binding strongly favors domain docking (*Marcinowski et al., 2011*). The findings presented here suggest that even when bound to ADP the equilibrium of AMPylated BiP is shifted towards a conformation in which the SBD and NBD are docked onto each other, thus resembling the ATP-bound state of unmodified BiP. This is evident both in solution-based protease-sensitivity experiments that track the disposition of the interdomain linker of BiP (a proxy for its conformational state, *Zhuravleva et al., 2012*) and in the crystal structure of AMPylated BiP, which is found in a domain-docked conformation despite the absence of ATP in the nucleotide binding domain (*Figure 7A*).

The substrate-binding domain of AMPylated apo BiP is found in a conformation typical of ATP-bound Hsp70/BiP (*Kityk et al., 2012*; *Yang et al., 2015*) and its widened substrate binding groove likely underlies the destabilizing effect of AMPylation on substrate binding. This structural feature is concordant with the enfeebling effect of AMPylation on BiP oligomerization in vitro (noted here), with previous observations that AMPylation correlates with an increased pool of monomeric cellular BiP and with higher substrate dissociation rates from ADP-bound AMPylated BiP observed in vitro (*Preissler et al., 2015b*', Figure 8E therein).

Loop $L_{7,8}$, comprising T518, and the connected β-sheet in the SBDβ subdomain undergo major rearrangements during the transition between the nucleotide-dependent conformations (*Yang et al., 2015*; *Zhuravleva and Gierasch, 2015*). In the ADP-bound state the T518 side chain is oriented inwards and contributes to contacts amongst other loop residues. These contacts are lost upon ATP binding (*Yang et al., 2015*), confining AMPylation to the ATP-bound conformation of BiP (*Preissler et al., 2015b*', Figure 7 therein). In the structures presented here the modified T518 side chain was flipped outward and pointed into the solvent. This suggests that the bulky and hydrophilic AMP moiety sterically prevents loop $L_{7,8}$ from reverting to its ADP-bound position. As a consequence, the connected downstream β-sheet$_8$ likely retains its conformation even in absence of ATP, biasing the SBD towards a conformation associated with high substrate 'off' rates (*Zhuravleva and Gierasch, 2015*) and stabilizing the contacts at the interface between the SBD and NBD, including the linker region. The net effect is to interfere with the allosteric transition to the domain-undocked conformation, normally observed in the ADP-bound state.

The benefit of inactivating BiP in a conformation that precludes stable engagement of substrates may arise in circumstances of an excess of BiP over its client proteins; for example, in the declining phase following a physiological burst of secreted protein synthesis. By interfering with the ability of excess BiP to engage substrates, AMPylation works against the tendency of the over-chaperoned ER to degrade clients that would otherwise be secreted (*Dorner et al., 1992*). Furthermore, over-expression of BiP that is locked in a conformation of slow substrate release is detrimental to ER function (*Hendershot et al., 1995*), suggesting how the emergence of a mechanism for inactivating excess BiP in the alternative conformation, specified by AMPylation, may have been selected.

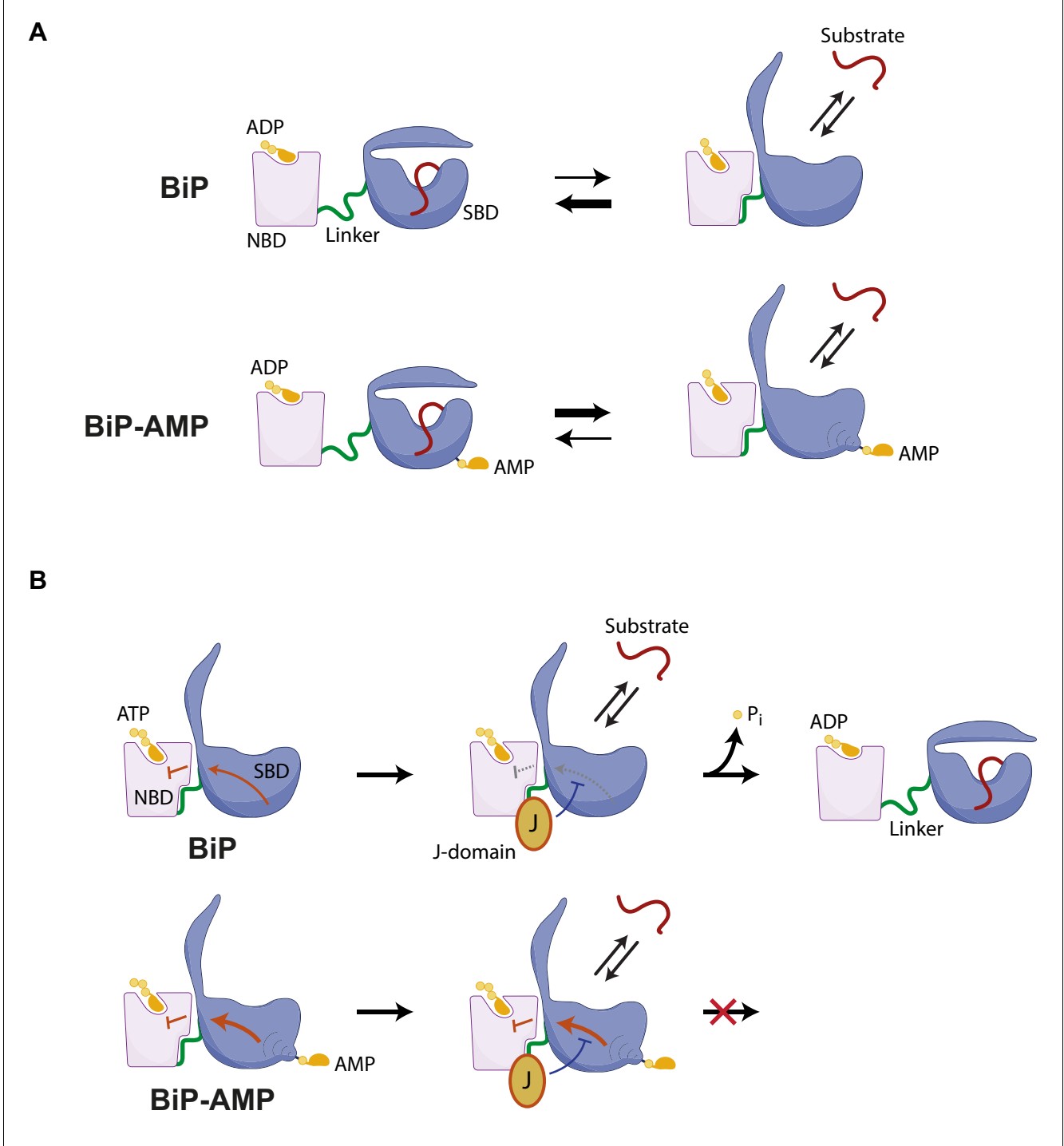

**Figure 7.** Hypothesized mechanisms by which AMPylation inactivates BiP. (**A**) ADP-bound unmodified BiP is strongly biased towards a domain-undocked conformation with low substrate 'off' rates. AMPylation (BiP-AMP) biases the ADP-bound chaperone towards a more domain-docked state with higher substrate 'off' rates. As a consequence, AMPylation enfeebles BiP-client interactions. (**B**) The contacts formed between the docked nucleotide binding domain (NBD) and substrate binding domain (SBD) of ATP-bound BiP are proposed to inhibit its basal ATPase activity (red arrow). J-domain interaction with BiP likely weakens these inhibitory interdomain contacts favoring ATP hydrolysis (*Kityk et al., 2015*) and triggering complete domain undocking, exposure of the interdomain linker (green), and stable substrate binding (red). By an allosteric mechanism, AMPylation on threonine 518 further strengthens the domain-docked conformation of ATP-bound BiP, which also strengthens the ATPase-inhibitory interdomain contacts (bold red arrow). This elevates the threshold for J domain-mediated stimulation of ATPase activity.
DOI: https://doi.org/10.7554/eLife.29428.024

## AMPylation interferes with BiP's responsiveness to J-proteins

The consequences of AMPylation to BiP's functional interactions with ER-localized J-proteins may be even more important than the effects of AMPylation on substrate binding in the ADP-bound state. In cells the interaction of Hsp70 proteins with their clients is governed by J-domain co-chaperones that instruct a non-equilibrium ATP hydrolysis-driven conformational cycle (*Kampinga and Craig, 2010*). The initial interactions between the ATP-bound chaperones and their substrates occur with high association rates followed by nucleotide hydrolysis that then strongly decreases the dissociation of the chaperone-substrate complex. Under physiological conditions (of excess ATP) J-proteins recruit ATP-bound Hsp70s to their substrates and thus couple substrate binding to acceleration of ATP hydrolysis, which increases the effective binding affinity beyond the equilibrium values observed in either nucleotide-bound state, thus establishing a regime of ultra-affinity (*De Los Rios and Barducci, 2014*).

AMPylation strongly interferes with the ability of J-proteins to impart ultra-affinity on BiP. This is reflected both by the dramatic defect in J protein-driven BiP oligomerization and by the defect in J protein-mediated BiP association with a model substrate in BLI experiments. Both the higher dissociation rate of substrates from ADP-bound AMPylated BiP and the defect in J protein-stimulated ATP hydrolysis appear to contribute to the inability of AMPylated BiP to attain ultra-affinity for its substrates. Thus, by targeting the rate-limiting step of its ATPase cycle AMPylation favors the retention of BiP in an inert, mostly ATP-bound conformation, whilst impeding wasteful ATP hydrolysis.

The significance of the modest, but reproducible defect observed in nucleotide exchange from AMPylated BiP, remains to be determined. The rate of ATP hydrolysis we observed was slower than ADP dissociation, suggesting that AMPylated BiP spends most of its time in the ATP-bound state (a feature that is likely accentuated in cells by the presence of nucleotide exchange factors). Nonetheless, a slowing of nucleotide exchange would serve to further limit ATP consumption by AMPylated BiP, whilst retaining even the small ADP-bound AMPylated fraction of BiP in a relatively inert state.

## Insights into AMPylation-mediated interference with J-protein stimulated ATP hydrolysis

AMPylation does not affect the ability of BiP to engage in protein-protein interactions with the J-domain - unmodified and AMPylated BiP both bind the J-domain with similar affinities - rather AMPylation blocks the subsequent stimulation of ATP hydrolysis. Can this be explained by the structural effect of the modification?

It has been proposed that the interdomain contacts formed between the SBD and NBD in the ATP-bound conformation of the *E. coli* Hsp70 DnaK communicate to the active site within the NBD to inhibit ATP hydrolysis (*Kityk et al., 2015*). Docking of the SBD against the NBD likely exerts a similar inhibitory effect in BiP, as the ATPase activity of the isolated NBD is 2-fold greater than that of intact BiP (*Hendershot et al., 1995*) and BiP's ATPase activity is increased 2-fold by mutations that interfere with domain docking (*Awad et al., 2008*). Furthermore, the SBD-NBD contacts identified by Kityk et al. (*Kityk et al., 2015*) as having a role in enforcing the inhibited state, are largely conserved in the crystal structure of AMPylated BiP.

J-proteins, which bind at the interface between the NBD and SBD, are proposed to weaken the interaction between the two, relieving the repression and thereby favoring ATP hydrolysis (*Awad et al., 2008*). Our findings suggest that BiP AMPylation further biases even the ATP-bound BiP towards the domain-docked conformation (compare reactions 4 and 8 in *Figure 1*). This likely arises from the presence of a bulky modification on T518, which prevents major rearrangements in the SBDβ by enforcing a specific conformation on loop $L_{7,8}$, thus stabilizing contacts between the SBD and NBD, frustrating J-protein action (*Figure 7B*). The importance of long-range allosteric communication between the SBD and NBD of BiP and the impact of AMPylation and other subtle changes in the SBD on this allostery receive independent support from NMR studies of the Zhuravleva lab (*Wieteska et al., 2017*). Though situated on the opposite end of the SBD, AMPylation allosterically targets the principal mechanism by which J-proteins enhance ATP hydrolysis by the NBD of Hsp70s.

## Materials and methods

### Plasmids

The plasmids used in this study were described previously or generated by standard molecular cloning techniques and are listed in *Supplementary file 2*.

### Protein purification

Wildtype and mutant Chinese hamster BiP proteins carrying an N-terminal hexahistidine (His6)-tag were purified as described previously (*Preissler et al., 2015b*) with modifications. Proteins were expressed in M15 *E. coli* cells (Qiagen, Hilden, Germany) and the bacterial cultures were grown at 37°C to an optical density ($OD_{600nm}$) of 0.8 in LB medium containing 50 µg/ml kanamycin and 100 µg/ml ampicillin. Protein expression was induced with 1 mM isopropylthio β-D-1-galactopyranoside (IPTG) and cells were further incubated at 37°C for 6 hr before they were harvested by centrifugation. The cells were lysed with a high-pressure homogenizer (EmulsiFlex-C3; Avestin, Mannheim, Germany) in buffer A [50 mM Tris-HCl pH 7.5, 500 mM NaCl, 1 mM $MgCl_2$, 0.2% (v/v) Triton X-100, 10% (v/v) glycerol, 20 mM imidazole] containing protease inhibitors [2 mM phenylmethylsulphonyl fluoride (PMSF), 4 µg/ml pepstatin, 4 µg/ml leupeptin, 8 µg/ml aprotinin] and 0.1 mg/ml DNaseI. The lysates were centrifuged for 30 min at 25,000 *g* and incubated with 1 ml Ni-NTA agarose (Qiagen) per 1 l of expression culture for 2 hr rotating at 4°C. The matrix was then transferred to a gravity-flow column and washed with buffer B [50 mM Tris-HCl pH 7.5, 500 mM NaCl, 0.2% (v/v) Triton X-100, 10% (v/v) glycerol, 30 mM imidazole] followed by buffer C [50 mM HEPES-KOH pH 7.4, 300 mM NaCl, 5% (v/v) glycerol, 10 mM imidazole, 5 mM β-mercaptoethanol] and further wash steps in buffer C supplemented sequentially with (i) 1% (v/v) Triton X-100, (ii) 1 M NaCl, (iii) 3 mM $Mg^{2+}$-ATP, or (iv) 0.5 M Tris-HCl pH 7.5. After a further wash step in buffer C containing 35 mM imidazole the retained BiP proteins were eluted with buffer D [50 mM HEPES-KOH pH 7.5, 300 mM NaCl, 5% (v/v) glycerol, 5 mM β-mercaptoethanol, 250 mM imidazole] and dialyzed against HKM buffer (50 mM HEPES-KOH pH 7.4, 150 mM KCl, 10 mM $MgCl_2$). The proteins were concentrated using centrifugal filters (Amicon Ultra, 30 kDa MWCO; Merck Millipore, Darmstadt, Germany), snap-frozen in liquid nitrogen, and stored at −80°C.

To generate nucleotide-free (apo) BiP preparations the purified proteins were further dialyzed twice for 12 hr at 4°C against buffer E (50 mM Tris-HCl pH 7.4, 150 mM NaCl, 5 mM EDTA), twice for 12 hr against buffer E containing 2 mM EDTA, once for 6 hr against buffer E without EDTA, and twice for 12 hr against HK buffer (50 mM HEPES-KOH pH 7.4, 150 mM KCl). The effectiveness of this treatment to remove bound nucleotide was confirmed by ion pair HPLC as described previously (*Preissler et al., 2017*). Nucleotide-free BiP proteins were used in the experiment described in *Figure 2—figure supplement 2*.

The *E. coli* Hsp70 protein DnaK (*Figure 6—figure supplement 1B*) was expressed as a fusion protein with an N-terminal His6-Smt3 from a pET24-based plasmid (UK 2243) in *E. coli* C3013 BL21 T7 Express *lysY/Iq* cells (New England BioLabs, Ipswich, MA). The cells were grown in LB medium containing 50 µg/ml kanamycin to $OD_{600nm}$ 0.6 at 37°C and expression was induced with 0.4 mM IPTG at 30°C for 4 hr. The cells were lysed and Ni affinity chromatography was performed as described above (see BiP purification), except that 1 mM β-mercaptoethanol was added to all buffers. For cleavage of His6-Smt3, Ulp1 enzyme was added to the eluted protein in a 1:1000 mass ratio together with 1 mM ATP and dialyzed for 16 hr at 4°C against HKM buffer supplemented with 1 mM β-mercaptoethanol. After addition of 2 mM ATP the dialyzed solution was immediately passed through a Superdex 200 10/300 GL gel filtration column (GE Healthcare, Chicago, IL) connected in series with a 1 ml HisTrap HP column (GE Healthcare) in HKM buffer containing 1 mM β-mercaptoethanol. The DnaK containing elution fractions were pooled, concentrated, and flash frozen in aliquots.

Human Grp170 protein (UK 1264) carrying an N-terminal His6-tag was expressed in *E. coli* BL21 (DE3) cells. The cells were grown at 37°C in LB medium containing 50 µg/ml kanamycin to $OD_{600nm}$ 0.6. The cells were then shifted to 20°C for 30 min and expression was induced with 1 mM IPTG for 4 hr. The cells were lysed and the protein was affinity purified as described above (see BiP purification) with the exception that detergent was omitted and 5 mM ATP was present in all solutions. The final wash steps of the affinity matrix were performed with buffer C containing 5 mM ATP sequentially

supplemented with (i) 0.5 M NaCl, (ii) 0.25 M Tris-HCl pH 7.5, or (iii) 35 mM imidazole. The bound protein was eluted with buffer D containing 5 mM ATP and further purified by size-exclusion chromatography using a Superdex 200 10/300 GL column (GE Healthcare) in HKM buffer containing 0.5 mM ATP. The protein eluted in two main peaks (the earlier one likely representing inactive oligomeric assemblies) and Grp170 containing fractions of the later eluting peak were pooled and immediately frozen in aliquots and stored at −80°C. SDS-PAGE analysis and Coomassie staining revealed that the final preparation still contained additional faster migrating protein species (*Figure 6—figure supplement 1A*). These likely comprise proteolytic fragments of full-length Grp170 as in-gel tryptic digest and mass spectrometry analysis of the corresponding gel bands identified mainly Grp170-derived peptides (*Figure 6—figure supplement 1A* and *Supplementary file 1*). Accordingly, most species efficiently bound to Ni-NTA agarose under denaturing conditions (*Figure 6—figure supplement 1A*). Although Grp170 is glycosylated in mammalian cells (*Lin et al., 1993*) bacterially expressed Grp170 stimulated MADA-ADP release from BiP in a concentration-dependent manner, but not from the bacterial Hsp70 DnaK, indicating its functionality and specificity in this assay (*Figure 6—figure supplement 1B*).

Expression and purification of N-terminally GST-tagged AMPylation-active GST-FICD$^{E234G}$ mutant protein was performed as described earlier (*Preissler et al., 2015b*). The protein was produced in *E. coli* C3013 BL21 T7 Express *lysY/Iq* cells at 37°C in LB medium containing 100 µg/ml ampicillin. Expression was induced at $OD_{600nm}$ 0.8 with 0.5 mM IPTG and the cultures were shifted to 20°C for 16 hr. The cells were harvested and lysed as described above in lysis buffer [50 mM Tris-HCl pH 7.5, 500 mM NaCl, 1 mM MgCl$_2$, 2 mM dithiothreitol (DTT), 0.2% (v/v) Triton X-100, 10% (v/v) glycerol] containing protease inhibitors and DNaseI. The lysate was centrifuged for 30 min at 25,000 *g* and incubated with 0.7 ml Glutathione Sepharose 4B (GE Healthcare) per 1 l of expression culture. After incubation for 2 hr at 4°C the beads were washed extensively with wash buffer F [50 mM Tris-HCl pH 7.5, 500 mM NaCl, 1 mM DTT, 0.2% (v/v) Triton X-100, 10% (v/v) glycerol] containing protease inhibitors, wash buffer G [50 mM Tris-HCl pH 7.5, 300 mM NaCl, 10 mM MgCl$_2$, 1 mM DTT, 0.1% (v/v) Triton X-100, 10% (v/v) glycerol] containing protease inhibitors, and wash buffer G sequentially supplemented with (i) 1% (v/v) Triton X-100, (ii) 1 M NaCl, (iii) 3 mM ATP, or (iv) 0.5 M Tris-HCl pH 7.5. Retained protein was eluted with elution buffer H [50 mM HEPES-KOH pH 7.4, 100 mM KCl, 4 mM MgCl$_2$, 1 mM CaCl$_2$, 0.1% (v/v) Triton X-100, 10% (v/v) glycerol, 40 mM reduced glutathione], snap-frozen in liquid nitrogen, and stored at −80°C.

The wildtype J-domain (UK 185) of mouse P58/ERdj6 (residues 384–470) and the H422Q mutant thereof (UK 186) were expressed as GST-fusion proteins (here referred to as GST-J$^{WT}$ and GST-J$^{QPD}$, respectively) according to (*Petrova et al., 2008*) and purified by gluthathione (GSH) affinity chromatography as described above. Proteins were eluted in buffer H containing 1 mM DTT, dialyzed over night against HKM at 4°C, and aliquots were frozen in liquid nitrogen and stored at −80°C. A fraction of purified GST-J$^{WT}$ or GST-J$^{QPD}$ proteins were adjusted to 160 µM and biotinylated with 1.65 mM Biotin-maleimide (Sigma, St. Louis, MO, cat. no. B1267) for 1 hr at 24°C in HKM buffer. The reaction was stopped on ice and proteins were passed through a Centri•Pure P25 desalting column (emp BIOTECH, Berlin, Germany) equilibrated with HKM. The protein containing elution fractions were pooled and aliquots were stored frozen.

## AMPylation of purified BiP proteins

AMPylation of purified BiP proteins was performed as previously described (*Preissler et al., 2015b*) with minor modifications. Purified BiP proteins were incubated for 6 hr at 30°C with 0.25 mg bacterially expressed GST-FICD$^{E234G}$ per 20 mg of BiP protein in presence of 3 mM ATP in buffer I [25 mM HEPES-KOH pH 7.4, 100 mM KCl, 10 mM MgCl$_2$, 1 mM CaCl$_2$, 0.1% (v/v) Triton X-100] followed by binding to Ni-NTA agarose beads for 1 hr at 25°C. The beads were washed with buffer I and eluted in buffer I containing 350 mM imidazole for 45 min at 25°C. The eluates were desalted using a Centri•Pure P25 column equilibrated in HKM buffer. The protein-containing fractions were pooled, frozen in liquid nitrogen, and stored at −80°C. Unmodified BiP prepared from mock AMPylation reactions without enzyme served as a control in the experiments. Intact protein mass spectrometry analysis of a representative preparation confirmed modification at high stoichiometry (*Figure 1—figure supplement 3*).

## Limited proteolysis experiments

For SubA-mediated BiP cleavage experiments purified wildtype or mutant BiP proteins were adjusted to a concentration of 0.5 µg/µl in HKM buffer containing 3 mM ATP, ADP or no added nucleotide. The reactions were incubated at 30°C and samples were taken before and at the indicated time intervals after addition of 20 ng/µl SubA protease. The withdrawn samples were immediately denatured in SDS-sample buffer and heated for 5 min at 75°C. The digested samples together with the undigested controls were analyzed by reducing SDS-PAGE and the proteins on the gels were visualized by Coomassie staining with InstantBlue solution (expedeon, Over, United Kingdom). The band intensities were quantified with Image J64 (NIH; RRID: SCR_003070).

## Protein crystallization and structure determination

Chinese hamster (*Cricetulus griseus*) BiP (residues 28–549) with a T229A mutation (*Figure 2A*) was expressed as a His6-Smt3 fusion protein (UK 1607) in M15 *E. coli* cells. The bacterial cultures were grown at 37°C to $OD_{600nm}$ 0.8 in LB medium containing 50 µg/ml kanamycin and 100 µg/ml ampicillin and expression was induced with 1 mM IPTG. The cells were further grown at 25°C for 14 hr, harvested and lysed in high-salt buffer (50 mM Tris-HCl pH 7.4, 500 mM NaCl, 1 mM $MgCl_2$, 2 mM PMSF) containing protease inhibitors and DNaseI as described above. The lysate was cleared by centrifugation for 30 min at 25,000 *g*, passed through a 0.45 µm syringe filter, and supplemented with 25 mM imidazole. The following purification strategies resulted in protein crystals that were suitable for X-ray data collection.

Strategy A (PDB 5O4P): The cleared lysate from a 4 l expression culture was passed over a 5 ml Ni-Sepharose HisTrap HP column (GE Healthcare) at 4°C. The column was washed sequentially with 30 ml high-salt buffer and 30 ml low-salt buffer (50 mM Tris, pH7.4, 100 mM NaCl, 1 mM $MgCl_2$) both containing 25 mM imidazole and 1 mM ATP. Bound protein was eluted with low-salt buffer supplemented with 500 mM imidazole (pH 7.4) followed immediately by addition of 10 mM ATP, 10 mM $MgCl_2$ and 1 mM Tris(2-carboxyethyl)phosphine (TCEP). Purified GST-FICD[E234G] protein and Ulp1 protease were added in a 60:1 and 1000:1 (His6-Smt3-haBiP:X) mass ratio, respectively, and incubated 14 hr at 24°C to simultaneously allow for AMPylation and cleavage of the His6-Smt3 tag. The solution was then supplemented with 100 mM EDTA and the protein was further purified by size-exclusion chromatography using a HiLoad 16/60 Superdex 75 prep grade column (GE Healthcare) equilibrated with GF buffer (5 mM HEPES-KOH pH 7.6, 100 mM NaCl, 0.1 mM EDTA, 0.1 mM TCEP). A Glutathione Sepharose 4B column (GSTrap 4B; GE Healthcare) was connected in series with the gel filtration column to retain GST-FICD[E234G] protein. The eluted protein was concentrated to >50 mg/ml in presence of 1 mM TCEP and crystallization was performed in 96-well sitting drop plates by combining 150 nl protein solution with 150 nl reservoir solution and equilibration at 20°C against 80 µl reservoir solution. Diffraction quality crystals grew in a solution containing 100 mM HEPES pH 7.5 and 1.5 M $Li_2SO_4$. Crystals were soaked in cryosolution [the precipitant solutions containing 20% (v/v) glycerol] and snap frozen in liquid nitrogen. Diffraction data were collected at the Diamond Light Source beamline I02 (or I24, Didcot, United Kingdom; see *Table 1*) and processed with Mosflm (*Battye et al., 2011*) and Aimless (*Evans, 2011*). The structure was solved by molecular replacement in Phaser (*McCoy et al., 2007*) by searching 2 copies of the nucleotide binding domain (PDB 3IUC) and substrate binding domain (PDB 4B9Q). Manual model building was carried out in COOT (*Emsley et al., 2010*) and further refinements in refmac5 (*Winn et al., 2001*) and phenix.refine (*Adams et al., 2010*). The final refinement statistics are summarized in *Table 1*. The structural graphs were generated with PyMOL software (PyMOL version 1.5.0.4; RRID: SCR_000305).

Strategy B (PDBs 6EOB and 6EOC): Protein was expressed in a 6 l culture and cell lysis was performed as described in strategy A. The lysate was supplemented with 0.1 mM TCEP and incubated with 6 ml Ni-NTA agarose for 1 hr while slowly rotating at 4°C. The suspended matrix was then transferred to a Glass Econo-column (2.5 × 10 cm; Bio-Rad, Hercules, CA) and washed first with high-salt buffer containing protease inhibitors and then with wash buffer J (50 mM Tris-HCl pH 8, 500 mM NaCl, 30 mM imidazole). Further wash steps were performed with wash buffer K (50 mM Tris-HCl pH 8, 300 mM NaCl, 10 mM imidazole, 5 mM β-mercaptoethanol) sequentially supplemented with (i) 1 M NaCl, (ii) 10 mM $MgCl_2$ and 3 mM ATP, (iii) 0.5 M Tris-HCl pH 8 and (iv) 40 mM imidazole. The retained protein was eluted with elution buffer L (50 mM Tris-HCl pH 7.4, 100 mM NaCl, 250 mM imidazole, 10 mM $MgCl_2$, 10 mM ATP, 1 mM TCEP). After AMPylation with GST-FICD[E234G] and cleavage of the His6-

Smt3 tag with Ulp1 (as described in strategy A) the protein solution was incubated with 1 ml Glutathione Sepharose 4B for 2 hr at room temperature (to bind GST-FICD$^{E234G}$). The matrix was collected by centrifugation and the supernatant was dialyzed twice for 14 hr against 5 l buffer M (25 mM Tris-HCl pH 7.4, 50 mM NaCl, 5 mM EDTA) and once for 24 hr against 10 l AEX-LS buffer (25 mM Tris-HCl pH 8, 50 mM NaCl). Afterwards, the protein was subjected to anion exchange chromatography using a 5 ml HiTrap Q HP column (GE Healthcare) equilibrated in AEX-LS buffer and bound protein was eluted with AEX-HS buffer (25 mM Tris-HCl pH 8, 1 M NaCl) on a linear gradient (0% to 50% AEX-HS in 20 column volumes). The purest fractions were pooled and incubated with 1.5 ml Ni-NTA agarose beads in presence of 25 mM imidazole for 30 min at 4°C (to bind residual uncleaved protein and free His6-Smt3 tag). To crystallize AMPylated BiP in absence of nucleotide the protein solution was desalted over a HiLoad 16/60 Superdex 75 prep grade column equilibrated with buffer N (10 mM HEPES-KOH pH 7.4, 100 mM NaCl). The protein solution was then supplemented with 1 mM TCEP and concentrated for crystallization. Crystals grown in 20% PEG-1000, 0.1 M NaKHPO$_4$ pH 6.2, 0.1 NaCl (PDB 6EOB) and 5% PEG-1000, 0.2 M Li$_2$SO$_4$, 0.1 M Na$_2$HPO$_4$ pH 4.2 (PDB 6EOC) were used for data collection and the structures were solved as described above.

Strategy C (PDB 6EOE): Protein was expressed, purified, and AMPylated as in strategy B. Protein obtained after ion exchange and reverse Ni affinity chromatography was incubated with 14 mM MgCl$_2$ and 10 mM ATP for 2 hr on ice before addition of 100 mM EDTA and immediate gel filtration using a HiLoad 16/60 Superdex 200 prep grade column (GE Healthcare) in buffer TN (25 mM Tris-HCl pH 8, 150 mM NaCl). Fractions of the main elution peak (corresponding to monomeric BiP protein) were pooled, supplemented with 1 mM TCEP, and concentrated for crystallization. The structure of BiP crystallized in 5% PEG-1000, 0.2 M Li$_2$SO$_4$, 0.1 M Na$_2$HPO$_4$ pH 4.2 was solved as described above.

Strategy D (PDB 6EOF): Protein was initially purified according to strategy B except that dialysis (after AMPylation and removal of the modifying enzyme) was performed against buffer O (25 mM Tris-HCl pH 7.4, 300 mM NaCl) for 16 hr at 4°C. The dialyzed solution was passed through a column containing 2 ml Q Sepharose High Performance matrix (GE Healthcare) by gravity flow. The flow-trough was concentrated and applied to gel filtration using a Superdex 200 10/300 GL column equilibrated in buffer TN. The purest elution fractions were pooled and dialyzed twice for 24 hr at 4°C against 5 l buffer TN containing 5 mM EDTA (to remove nucleotides) and once against the same buffer without EDTA. The protein solution was then supplemented with 1 mM TCEP and concentrated. The crystallization reactions were set up in presence of 10 mM ADP and 14 mM MgCl$_2$. Crystals grown in 9% PEG-1000, 0.2 M Li$_2$SO$_4$, 0.1 M Na$_2$HPO$_4$ pH 4.4 were used for data collection and the structure was solved as described above.

## Analytical size-exclusion chromatography

Analytical size-exclusion chromatography (SEC) was performed as described previously (*Preissler et al., 2015a*) with modifications. Purified BiP proteins were adjusted to 50 µM in HKM buffer and incubated in a final volume of 25 µl at room temperature for 20 min before injection. Where indicated ADP or ATP (each at 1.5 mM) as well as GST-J$^{WT}$ or GST-J$^{QPD}$ J-domain proteins (each between 1.25 µM and 10 µM) were added to the reactions. 10 µl of each sample were injected onto a SEC-3 HPLC column (300 Å pore size; Agilent Technologies, Santa Clara, CA) equilibrated with HKM and the runs were performed at a flow rate of 0.3 ml/min at room temperature. Peptide bond absorbance at 230 nm (A$_{230 \, nm}$) was detected and plotted against the elution time. A gel filtration standard (Bio-Rad, cat. no. 151–1901) was applied as a size reference and the elution peaks of Thyroglobulin (670 kDa), γ-globulin (158 kDa), Ovalbumin (44 kDa), and Myoglobulin (17 kDa) are indicated.

## Bio-layer interferometry

Bio-layer interferometry (BLI) experiments were performed on the FortéBio Octet RED96 System (Pall FortéBio, Menlo Park, CA). To study J domain-dependent substrate interactions (*Figure 4C*, *Figure 4—figure supplement 1* and *Figure 5—figure supplement 2B*) unmodified or in vitro AMPylated BiP proteins were diluted to 15 µM in HKM buffer containing 0.05% (v/v) Triton X-100. Biotinylated GST-J$^{WT}$ and GST-J$^{QPD}$ were diluted to 20 nM and P15 peptide (ALLLSAPRRGAGKK; custom synthesized by GenScript, Piscataway, NJ), which was biotinylated on the C-terminal lysine, was diluted to 50 nM in the same solution. Streptavidin (SA)-coated biosensors (Pall FortéBio) were

hydrated in assay solution for at least five minutes before the experiment. Where indicated 2 mM ADP or ATP was added. All solutions were prepared in a final volume of 200 µl on a 96 well microplate (greiner bio-one, Kremsmünster, Austria, cat. no. 655209) and data acquisition was performed at a shake speed of 600 rpm and at 24°C with the indicated experimental steps. In brief, after an initial equilibration step in assay solution biotinylated GST-J$^{WT}$ or GST-J$^{QPD}$ was immobilized until a difference in the binding signal of 0.4 nm was reached. The sensors were then washed and saturated with P15 peptide. Following a baseline step the sensors were introduced in BiP protein-containing solution to measure association followed by dipping into protein-free solution to record BiP dissociation. The dissociation data from at least three independent repeats were fitted with a one-phase decay function to determine the dissociation rate constants using the GraphPad Prism 6 software (GraphPad Software; RRID: SCR_002798).

The experiments to analyze non-substrate interactions between the J-domain and BiP (*Figure 5*, *Figure 5—figure supplement 1* and *Figure 5—figure supplement 2A*) were performed likewise without P15 peptide and the sensors were saturated with biotinylated GST-J$^{WT}$ or GST-J$^{QPD}$ (both at 20 nM) before exposure to unmodified or AMPylated BiP$^{T229A-V461F}$ (UK 1825) at the indicated concentrations. During the baseline, association, and dissociation steps 1 mM ATP was present. The dissociation constants were determined by plotting the steady state binding signals against the BiP concentration and fitting the data with GraphPad Prism 6.

## Single-turnover ATPase assay

ATPase assays under single-turnover conditions were performed as described previously (*Mayer et al., 1999*) with modifications. Unmodified and AMPylated BiP proteins were adjusted to 30 µM in reaction solution (25 mM HEPES-KOH pH 7.4, 50 mM KCl, 10 mM MgCl$_2$) and incubated in a final volume of 50 µl with 0.8 mM ATP and 0.6 MBq α-$^{32}$P-ATP (EasyTide; Perkin Elmer, Waltham, MA) for 3 min on ice. The formed BiP•ATP complexes were separated from free nucleotide by gel filtration using illustra Sephadex G-50 NICK columns (GE Healthcare, cat. no. 17-0855-01) that have been presaturated with 1 ml of a 1 mg/ml bovine serum albumin (BSA) solution and equilibrated with ice-cold reaction solution. Individual elution fractions (two drops each) were collected and the fractions of the first radioactive peak were pooled and flash frozen in aliquots. The final protein concentration of the pools was estimated based on the dilution of the sample during separation. For each reaction a fresh aliquot of the pre-formed complexes was rapidly thawed at room temperature. After 0.5 µl were withdrawn as a zero time point sample, 6 µl of the remaining BiP•ATP complexes were added to reactions containing GST-J$^{QPD}$ or GST-J$^{WT}$ proteins (both at 2 µM) in a final volume of 50 µl. The final BiP protein concentration in the reactions was ~0.8 µM. At the indicated time points 2 µl were withdrawn from the reactions and spotted onto a thin layer chromatography (TLC) plate (PEI Cellulose F; Merck Millipore, cat. no. 105579), which was pre-spotted with a mixture of 5 mM ADP and 5 mM ATP. The nucleotides were then separated by developing the TLC plates with 400 mM LiCl and 10% (v/v) acetic acid as a mobile phase. The plates were dried, exposed to a storage phosphor screen, and the signals were detected using a Typhoon Trio imager (GE Healthcare). The signals were quantified using Image J64 and the ADP values were normalized to the total radioactive signal (ADP + ATP) for each time point and fitted with a single exponential function using the GraphPad Prism 6 software.

## ADP release experiments

Non-AMPylated or AMPylated BiP proteins at 1 µM were incubated with 1 µM of the fluorescent ADP analog MABA-ADP (8-[(4-Amino)butyl]-amino-ADP - MANT; Jena Bioscience, Jena, Germany, cat. no. NU-893-MNT) (*Theyssen et al., 1996*) in HKM buffer for 2 hr at 30°C. Afterwards, the samples were mixed at 25°C in a 1:1 (v/v) ratio with HKM containing 2 mM ATP using a stopped-flow reaction analyzer (SX.18MV; Applied Photophysics, Leatherhead, United Kingdom). The fluorophore was excited at 360 nm and a 420 nm cut-off filter was used to detect emission. The traces of at least five consecutive injections per sample were averaged and the data were fitted with a single exponential decay function.

Stimulated MABA-ADP release (*Figure 6C* and *Figure 6—figure supplement 1B and C*) was measured in HKM containing 2 mM CaCl$_2$. For that, 2.5 µM BiP protein was incubated with 2.5 µM MABA-ADP for 2 hr at 30°C and then mixed in a 1:1 ratio with the same buffer solution containing 3 mM ATP without or with 1.4 µM Grp170. The decrease in the fluorescent signal was detected at

22°C in a 3 mm quartz cell (Hellma Analytics, Müllheim, Germany) with a fluorescence spectrometer (LS 55; PerkinElmer) at an excitation wavelength of 360 nm and emission was recorded at 440 nm. The dissociation rate constants were calculated by fitting a single exponential decay function to the data using GraphPad Prism 6.

## Mass spectrometry

Intact protein mass spectrometry was performed to analyze the AMPylation status of modified BiP protein used for biochemical experiments (*Figure 1—figure supplement 3*), to set up crystallization reactions, and of protein from dissolved crystals (*Figure 2—figure supplement 4*). For the latter, several remaining crystals grown under the condition that yielded structure PDB 6EOE (after having taken out the one used for X-ray data collection) were washed by dipping them sequentially three times in water and dissolving in 0.5 M sodium acetate. Liquid chromatography-mass spectrometry (LC–MS) was performed on a Xevo G2-S Tof mass spectrometer coupled to an ACQUITY UPLC system (Waters, Elstree, United Kingdom) using an ACQUITY UPLC BEH300 C4 column (1.7 μm, 2.1 × 50 mm). Solvents A (water with 0.1% formic acid) and B (95% acetonitrile, 4.9% water and 0.1% formic acid) were used as the mobile phase at a flow rate of 0.2 ml/min. The electrospray source was operated with a capillary voltage of 2.0 kV and a cone voltage of 40 V. Nitrogen was used as the desolvation gas at a total flow of 850 l/hr. Total mass spectra were reconstructed from the ion series using the MaxEnt algorithm preinstalled on MassLynx software (v. 4.1 from Waters; RRID: SCR_014271) according to the manufacturer's instructions.

For peptide mass spectrometry (*Figure 6—figure supplement 1A* and *Supplementary file 1*) selected protein bands were excised from an SDS-polyacrylamide gel and subjected to in-gel tryptic digestion. The resulting peptides were analyzed using a Q Exactive (Thermo Scientific, Waltham, MA) coupled to an RSLC3000nano UPLC (Thermo Scientific). Files were searched against a SwissProt database (downloaded 16/06/16, 551,385 entries) using Mascot 2.3 with peptide and protein validation performed in Scaffold Proteome Software 4.3.2 (RRID: SCR_014345).

## Affinity-tag based co-purification

The GSH pull-down experiment (*Figure 4A*) was performed as described previously (*Petrova et al., 2008*). For each reaction 15 μl Glutathione Sepharose 4B beads were incubated with 30 μg GST-J$^{WT}$ or GST-J$^{QPD}$ protein in HKM buffer containing 2 mM DTT for 30 min at room temperature while slowly rotating. The beads were collected by centrifugation for 2 min at 100 $g$ and washed twice in buffer P [20 mM HEPES-KOH pH 7.4, 75 mM KCl, 10 mM MgCl$_2$, 0.01% (v/v) Tween 20] to remove unbound protein followed by incubation with 30 μg unmodified or AMPylated BiP$^{WT}$ or BiP$^{V461F}$ proteins in buffer P supplemented with 3 mM ADP or ATP (as indicated) for 1 hr at 4°C. Afterwards, the beads were washed four times on ice with buffer P containing 1 mM ADP or ATP and bound proteins were eluted with 35 μl 2x SDS sample buffer for 5 min at 75°C. The samples were analyzed by reducing SDS-PAGE and Coomassie staining. Samples of the adjusted BiP protein solutions were loaded as an input control.

For the Ni-NTA pull-down experiment (*Figure 6—figure supplement 1A*) to identify N-terminal His6-tagged fragments of purified Grp170 (UK 1264) 10 μg protein per sample was incubated for 16 hr at 4°C with 30 μl Ni-NTA agarose beads in HKM buffer containing 2 mM β-mercaptoethanol and 20 mM imidazole either under native or denaturing [with 6 M guanidine hydrochloride (GdnHCl)] conditions. The beads were then recovered by centrifugation for 5 min at 500 $g$ and washed with HKM (native sample) or HKM containing 4 M GdnHCl (denatured sample). After three further wash steps with HKM the bound proteins were eluted with SDS sample buffer containing 250 mM imidazole for 5 min at 75°C. Input samples and eluted proteins were analyzed by reducing SDS-PAGE and Coomassie staining.

## Statistics

Two-tailed unpaired $t$-tests were performed using GraphPad Prism 6.

## Acknowledgements

We thank Cláudia Rato (Ron lab) for advice and revising the manuscript as well as Robin Antrobus (CIMR, University of Cambridge, UK) and Dijana Matak-Vinkovic (Department of Chemistry,

University of Cambridge, UK) and their facility teams for mass spectrometry analyses. Further we thank Chris Johnson (Laboratory of Molecular Biology, Cambridge, UK) for technical support and access to the stopped-flow setup, Osamu Hori (Kanazawa University, Japan) for his gift of human Grp170 cDNA, the Huntington lab (CIMR, University of Cambridge, UK) for access to the Octet machine, as well as Anastasia Zhuravleva and Lukasz Wieteska (University of Leeds, UK) for exchange of unpublished data. Supported by Wellcome Trust Principal Research Fellowships to DR (Wellcome 200848/Z/16/Z) and RJR (Wellcome 082961/Z/07/Z), a grant from the British Heart Foundation (PG/ 12/41/29679) to YY, and a Wellcome Trust Strategic Award to the Cambridge Institute for Medical Research (Wellcome 100140).

## Additional information

### Competing interests

David Ron: Reviewing editor, *eLife*. The other authors declare that no competing interests exist.

### Funding

| Funder | Grant reference number | Author |
|---|---|---|
| Wellcome | Wellcome 200848/Z/16/Z | David Ron |
| Wellcome | Wellcome 082961/Z/07/Z | Randy J Read |
| Wellcome | Wellcome 100140 | David Ron |
| British Heart Foundation | PG/12/41/29679 | Yahui Yan |

The funders had no role in study design, data collection and interpretation, or the decision to submit the work for publication.

### Author contributions

Steffen Preissler, Conceived and led the project, Designed and conducted most experiments, Crystallized BiP, Analyzed and interpreted the data, Drafted the manuscript; Lukas Rohland, Conducted, analyzed, and interpreted experiments to study non-substrate BiP-J interactions, Contributed to protein purification, Revised the manuscript; Yahui Yan, Collected, processed, and analyzed the X-ray diffraction data, Contributed to interpretation of the structural data and figure preparation; Ruming Chen, Crystallized BiP and collected X-ray diffraction data, Contributed to X-ray data analysis and interpretation; Randy J Read, Contributed to analysis and interpretation of the structural data, Revised the manuscript; David Ron, Oversaw the project conception and design, Constructed plasmid DNA, Interpreted the data, Drafted and revised the manuscript

### Author ORCIDs

Steffen Preissler https://orcid.org/0000-0001-7936-9836
Lukas Rohland https://orcid.org/0000-0003-1559-5097
Yahui Yan https://orcid.org/0000-0001-6934-9874
Randy J Read https://orcid.org/0000-0001-8273-0047
David Ron https://orcid.org/0000-0002-3014-5636

### Decision letter and Author response

Decision letter https://doi.org/10.7554/eLife.29428.038
Author response https://doi.org/10.7554/eLife.29428.039

## Additional files

### Supplementary files

• Supplementary file 1. Mass spectrometry analysis of the Grp170 protein preparation. Bands of full-length Grp170 protein and faster migrating species were cut out of a Coomassie-stained SDS-PAGE

gel and analyzed by LC-MS/MS after in-gel digest with trypsin. The identified peptides for the bands indicated in *Figure 6—figure supplement 1A* are presented.

DOI: https://doi.org/10.7554/eLife.29428.025

• Supplementary file 2. Plasmids used in this study. The table lists the plasmids used in this study including references (as PMIDs).

DOI: https://doi.org/10.7554/eLife.29428.026

• Transparent reporting form

DOI: https://doi.org/10.7554/eLife.29428.027

## Major datasets

The following datasets were generated:

| Author(s) | Year | Dataset title | Dataset URL | Database, license, and accessibility information |
|---|---|---|---|---|
| Yahui Yan, Ruming Chen, David Ron, Randy J Read | 2017 | Crystal structure of AMPylated GRP78 | https://www.rcsb.org/pdb/explore/explore.do?structureId=5O4P | Publicly available at the RCSB Protein Data Bank (Accession no: 5O4P) |
| Yahui Yan, Steffen Preissler, David Ron, Randy J Read | 2017 | Crystal structure of AMPylated GRP78 in apo form (Crystal form 1) | https://www.rcsb.org/pdb/explore/explore.do?structureId=6EOB | Publicly available at the RCSB Protein Data Bank (Accession no: 6EOB) |
| Yahui Yan, Steffen Preissler, David Ron, Randy J Read | 2017 | Crystal structure of AMPylated GRP78 in apo form (Crystal form 2) | https://www.rcsb.org/pdb/explore/explore.do?structureId=6eoc | Publicly available at the RCSB Protein Data Bank (Accession no: 6EOC) |
| Yahui Yan, Steffen Preissler, David Ron, Randy J Read | 2017 | Crystal structure of AMPylated GRP78 with nucleotide | https://www.rcsb.org/pdb/explore/explore.do?structureId=6EOE | Publicly available at the RCSB Protein Data Bank (Accession no: 6EOE) |
| Yahui Yan, Steffen Preissler, Randy J Read, David Ron | 2017 | Crystal structure of AMPylated GRP78 in ADP state | https://www.rcsb.org/pdb/explore/explore.do?structureId=6EOF | Publicly available at the RCSB Protein Data Bank (Accession no: 6EOF) |

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
