## [Decision Letter]

Thank you for submitting your article "AMPylation targets the rate-limiting step of BiP's ATPase cycle for its functional inactivation" for consideration by *eLife*. Your article has been reviewed by two peer reviewers, and the evaluation has been overseen by a Reviewing Editor and Randy Schekman as the Senior Editor.

The reviewers have discussed the reviews with one another and the Reviewing Editor has drafted this decision to help you prepare a revised submission.

Summary of the work:

AMPylation inactivates the ER Hsp70 chaperone BiP. This study provides an explanation for the effect of AMPylation on BiP function, building on the earlier discovery of this modification. The authors show that AMPylation reduces the ATPase rate of BiP. Interestingly, J-protein binds to modified BiP but does not stimulate its ATPase. Crystallization of a construct of BiP that harbors the AMP modification shows that AMPylated BiP adopts the typical ATP-bound conformation in which the N-terminal ATPase domain and the C-terminal peptide binding domain are docked. The structure supports the effect of the AMP modification to shift the conformation of BiP towards the domain-docked state.

Essential points to be addressed in the revised manuscript:

1) In the crystal structure, AMP is not visible. The evidence for the presence of the modification is only indirect. The mass spec analysis was performed on the non-crystallized protein, but should be done on protein from a dissolved crystal. Also, it should be shown that the modified and unmodified forms behave similarly in the mass spec analysis. A 50:50 mixture of both forms could be analyzed.

2) ADP release was measured in the presence of Grp170. The protein shows impurities in the range of 75 and 60 kDa. The larger impurity could be DnaK, which might influence the results in the ADP release assay. The impurities should be analyzed by mass spectrometry.

3) Grp170 was produced in *E. coli*. This protein is known to be heavily glycosylated. Authors should comment on the functionality of the *E. coli* produced protein.

4) How reproducible are the proteolysis data? Corresponding panels in Figure 1 and Figure 2—figure supplement 2 for unmodified BiP, with the V461F mutation in ADP seem to show different amounts of residual full-length protein. Please comment on the reproducibility of this assay.

Other points:

1) The Abstract doesn't do justice to the story. Authors should consider making a clearer statement of the conclusions as was done in the Discussion.

2) The authors make their arguments about the docked and undocked allosteric equilibrium based on the behavior of DnaK. However, BiP may be different. Authors should at least state that they are assuming a similar behavior before they start talking about BiP in terms that have been developed with lots of data from DnaK behavior.

3) In the crystal structure a large part of the substrate binding domain (lid) is missing. The lid is known to be important for substrate binding and stimulation of ATP hydrolysis. This should be discussed in more detail.

---

## [Author Response]

Essential points to be addressed in the revised manuscript:1) In the crystal structure, AMP is not visible. The evidence for the presence of the modification is only indirect. The mass spec analysis was performed on the non-crystallized protein, but should be done on protein from a dissolved crystal. Also, it should be shown that the modified and unmodified forms behave similarly in the mass spec analysis. A 50:50 mixture of both forms could be analyzed.

The original crystals that gave rise to the data set in PDB 5O4P were no longer available for analysis. Therefore, we prepared new crystals of apo BiP, solved their structure (new Figure 2—figure supplement 3) and performed intact protein mass spectrometry that revealed the presence of a homogenous preparation of AMPylated BiP in the crystal (new Figure 2—figure supplement 4).

2) ADP release was measured in the presence of Grp170. The protein shows impurities in the range of 75 and 60 kDa. The larger impurity could be DnaK, which might influence the results in the ADP release assay. The impurities should be analyzed by mass spectrometry.

New Figure 6—figure supplement 1 and new Supplementary file 1 present an analysis of the Grp170 preparation used in our experiments.

Panel A shows that both the major species corresponding to full-length Grp170 and the unexpected species co-migrating with the 75 Kd marker are recovered by Ni-chelate chromatography under both native and denaturing conditions (6 M guanidinium HCl). It follows therefore that both possess a His-6 tag and argues that contaminating *E. coli* DnaK is unlikely to make an important contribution to the mass of the 75 KDa species. This impression is supported by mass spectrometric analysis of a tryptic digest of the 75 KDa species present in the Grp170 preparation, which shows that most of peptides recovered are from Grp170 and not from *E. coli* DnaK (Supplementary file 1).

To address head on the potential contribution of DnaK to the ADP-release data, we examined directly that potential role of Grp170 in stimulating ADP release from DnaK and found no evidence for such activity (panel B).

From these experiments we conclude that DnaK is not an important contributor to the Grp170 preparation, neither in content nor in functionality.

3) Grp170 was produced in E. coli. This protein is known to be heavily glycosylated. Authors should comment on the functionality of the E. coli produced protein.

We address this point in the fourth paragraph of the subsection “Protein purification”.

4) How reproducible are the proteolysis data? Corresponding panels in Figure 1 and Figure 2—figure supplement 2 for unmodified BiP, with the V461F mutation in ADP seem to show different amounts of residual full-length protein. Please comment on the reproducibility of this assay.

To address this issue we present three complete data sets of the entire experiment: The original Figure 1 (which remains unchanged) and two new data sets from experiments performed on different days (new Figure 1—figure supplement 2). We also provide a quantitative analysis of the variation observed across these three experiments (new Figure 1—figure supplement 2). Finally we present intact protein mass spectrometric data to confirm the completeness of AMPylation of the BiP used in the experiments. We believe the digestion patterns of the various BiP proteins reported here under diverse conditions to be highly reproducible.

Other points:1) The Abstract doesn't do justice to the story. Authors should consider making a clearer statement of the conclusions as was done in the Discussion.

We have revised the Abstract to explain more clearly the conclusions derived from this study.

2) The authors make their arguments about the docked and undocked allosteric equilibrium based on the behavior of DnaK. However, BiP may be different. Authors should at least state that they are assuming a similar behavior before they start talking about BiP in terms that have been developed with lots of data from DnaK behavior.

In the first paragraph of the subsection “AMPylation biases BiP towards a conformation normally attained by the ATP-bound chaperone”, we lay bare our assumptions in regards to the conservation in mechanism of DnaK, BiP and other Hsp70s.

3) In the crystal structure a large part of the substrate binding domain (lid) is missing. The lid is known to be important for substrate binding and stimulation of ATP hydrolysis. This should be discussed in more detail.

The first paragraph of the subsection “Structure of AMPylated BiP” discusses the potential impact of lid truncation.